# Distributed Quasi-Newton Method for Fair and Fast Federated Learning

**Shayan Mohajer Hamidi**                                        *smohajer@uwaterloo.ca*
*Department of Electrical & Computer Engineering*
*University of Waterloo*

**Linfeng Ye**                                        *linfeng.ye@mail.utoronto.ca*
*Edward S. Rogers Sr. Department of Electrical & Computer Engineering*
*University of Toronto*

**Reviewed on OpenReview:** *https://openreview.net/forum?id=KbteA50cni*

## Abstract

Federated learning (FL) is a promising technology that enables edge devices/clients to collaboratively and iteratively train a machine learning model under the coordination of a central server. The most common approach to FL is first-order methods, where clients send their local gradients to the server in each iteration. However, these methods often suffer from slow convergence rates. As a remedy, second-order methods, such as quasi-Newton, can be employed in FL to accelerate its convergence. Unfortunately, similarly to the first-order FL methods, the application of second-order methods in FL can lead to unfair models, achieving high average accuracy while performing poorly on certain clients' local datasets. To tackle this issue, in this paper we introduce a novel second-order FL framework, dubbed **d**istributed **q**uasi-**N**ewton **fed**erated learning (DQN-Fed). This approach seeks to ensure fairness while leveraging the fast convergence properties of quasi-Newton methods in the FL context. Specifically, DQN-Fed helps the server update the global model in such a way that (i) all local loss functions decrease to promote fairness, and (ii) the rate of change in local loss functions aligns with that of the quasi-Newton method. We prove the convergence of DQN-Fed and demonstrate its *linear-quadratic* convergence rate. Moreover, we validate the efficacy of DQN-Fed across a range of federated datasets, showing that it surpasses state-of-the-art fair FL methods in fairness, average accuracy and convergence speed. The Code for paper is publicly available at `https://anonymous.4open.science/r/DQN-Fed-FDD2`.

## 1 Introduction

Traditionally, machine learning (ML) models are trained centrally, with data stored in a central server. However, in modern applications, devices often resist sharing private data remotely. To address this, federated learning (FL) was introduced by McMahan et al. (2017), where each device trains locally with a central server. In FL, devices share only local updates, maintaining data privacy. FedAvg, proposed by McMahan et al. (2017), is a popular first-order FL method. It combines local stochastic gradient descent (SGD) on each client with iterative model averaging. The server sends the global model to selected clients (Eichner et al., 2019; Wang et al., 2021a), which perform local SGD on their training data. Local gradients are sent back to the server, which calculates their (weighted) average to update the global model iteratively.

Nevertheless, first-order FL methods tend to exhibit slow convergence, particularly in terms of the number of iterations or communication rounds required (Krouka et al., 2022). More precisely, the convergence rate of first-order FL algorithms is sublinear, i.e., the required number of communication rounds $T_\epsilon$ to achieve $\epsilon$-accurate solution is $T_\epsilon = \mathcal{O}(\frac{1}{\epsilon})$. Additionally, their convergence speed is highly influenced by the condition number, which is dependent on several factors, including: (i) the architecture of the model being trained, (ii) the choice of loss function, and (iii) the distribution of the training data (Elgabli et al., 2022).

To overcome this limitation, second-order methods can be applied in FL to significantly boost convergence speed (Safaryan et al., 2022; Elgabli et al., 2022; Hamidi et al., 2024a). By estimating the local curvature of the loss landscape, these methods provide more adaptive and efficient update directions, leading to faster and more reliable convergence (Battiti, 1992). Specifically, in second-order FL methods, the clients compute the Newton direction for their respective local loss functions and send these directions to the server. The server then averages the Newton directions from all clients and updates the global model in the direction of this average (Ghosh et al., 2020; Zhang & Lin, 2015). Moreover, since Newton's methods require calculating the inverse of the Hessian matrix at each iteration—a computationally expensive operation—the inverse is typically approximated using iterative techniques, leading to quasi-Newton methods (Wang et al., 2018).

While Newton-type methods accelerate the convergence of FL algorithms, they do not guarantee that the averaged Newton direction computed by the server is a descent direction for all clients—a limitation also present in first-order FL methods (Hu et al., 2022; Pan et al., 2024; Chen et al., 2024). In other words, upon updating the global model toward this averaged direction, the loss function for some client my not decrease, potentially leading to poor performance on their private datasets. As a result, the learned model might exhibit *unfairness*, with high average accuracy but poor performance for clients whose data distributions differ from the majority[1]. Thus, naively applying Newton methods in FL can lead to the training of an *unfair* model (see Section 5 for results).

To tackle the issue mentioned above, this paper presents a novel second-order FL framework, dubbed **d**istributed **q**uasi-**N**ewton **fed**erated learning (DQN-Fed). This approach aims to ensure fairness while leveraging the fast convergence properties of quasi-Newton methods in the FL setting. In particular, DQN-Fed is designed to assist the server in updating the global model such that (i) all local loss functions decrease resulting in training a *fair* model, and (ii) the rate of change in local loss functions aligns with the rate of change in the quasi-Newton method. To achieve this, based on the received local quasi-Newton directions and the local gradients, the server identifies an updating direction that satisfies both of the aforementioned conditions. This will in turn yield a *fair* FL algorithm, as the global updating direction is descent for all the clients. Moreover, the convergence of DQN-Fed is fast, as the rate of change in the local loss functions follow quasi-Newton methods.

In summary, the contributions of the paper are as follows:

• We introduce distributed quasi-Newton federated learning (DQN-Fed), a method designed to assist the server in updating the global model to achieve both fairness and fast convergence in FL.

• We present a closed-form solution for calculating the global updating direction, distinguishing our approach from many existing fair FL methods that depend on iterative or generic quadratic programming techniques.

• Leveraging common assumptions in FL literature, we establish the convergence proof for DQN-Fed algorithm across various FL setups. In addition, we prove the convergence rate of the proposed method, and show that DQN-Fed exhibits a linear-quadratic convergence rate. Specifically, the convergence is either quadratic, with $T_\epsilon = \mathcal{O}\left(\log\log\frac{1}{\epsilon}\right)$, or linear, with $T_\epsilon = \mathcal{O}\left(\frac{1}{\log(\frac{\lambda}{L\delta})}\log\frac{1}{\epsilon}\right)$, where $\lambda$, $L$ and $\delta$ are constants.

• Through *comprehensive* experiments conducted on **seven** different datasets (six vision datasets and one language dataset), we demonstrate that DQN-Fed attains superior fairness level among clients, and converges faster compared to the state-of-the-art fair alternatives.

## 2 Related Works

• **Fairness in FL.** The literature offers a myriad of perspectives to address the challenge of fairness in FL. These methods include client selection (Nishio & Yonetani, 2019; Huang et al., 2020a; 2022; Yang et al., 2021), contribution Evaluation (Zhang et al., 2020; Lyu et al., 2020; Song et al., 2021; Le et al., 2021), incentive mechanisms (Zhang et al., 2021; Kang et al., 2019; Ye et al., 2020; Zhang et al., 2020), and the methods based on the loss function. Specifically, our work falls into the latter category. This approach aims

---

[1]Learning an unfair model is a common challenge in first-order FL methods as well, and there is a substantial body of research dedicated to developing fair FL models (Mohri et al., 2019; Du et al., 2021; Li et al., 2020; Hu et al., 2022; Hamidi & YANG, 2024).

to achieve uniform test accuracy across clients. In particular, works within this framework focus on reducing the variance of test accuracy among participating clients. We provide a thorough review on fairness issue in ML and FL in Appendix J.

• **Second-Order FL methods.** DistributedNewton (Ghosh et al., 2020) and LocalNewton (Gupta et al., 2021) perform Newton's method instead of SGD on local machines to accelerate the convergence of local models. FedNew (Elgabli et al., 2022) utilized one pass ADMM on local machines to calculating local directions and approximate Newton method to update the global model. FedNL (Safaryan et al., 2022) send the compressed local Hessian updates to global server and performed Newton step globally. Based on eigendecomposition of the local Hessian matrices, SHED (Dal Fabbro et al., 2024) incrementally updated eigenvector-eigenvalue pairs to the global server and recovered the Hessian to use Newton method. Recently, Li et al. (2024) proposed federated Newton sketch methods (FedNS) to approximate the centralized Newton's method by communicating the sketched square-root Hessian instead of the exact Hessian.

## 3 Notation and Preliminaries

### 3.1 Notation

We denote by $[K]$ the set of integers $\{1, 2, \cdots, K\}$. In addition, we define $\{f_k\}_{k\in[K]} = \{f_1, f_2, \ldots, f_K\}$ for a scalar/function $f$. We use bold small letters to represent vectors, and bold capital letters to represent matrices. Denote by $\mathfrak{u}_i$ the $i$-th element of vector $\mathfrak{u}$. For two vectors $\mathfrak{u}, \mathfrak{v} \in \mathbb{R}^d$, we say $\mathfrak{u} \leq \mathfrak{v}$ iff $\mathfrak{u}_i \leq \mathfrak{v}_i$ for $\forall i \in [d]$. Denote by $\mathfrak{v} \cdot \mathfrak{u}$ their inner product, and by $\text{proj}_{\mathfrak{u}}(\mathfrak{v}) = \frac{\mathfrak{v} \cdot \mathfrak{u}}{\mathfrak{u} \cdot \mathfrak{u}} \mathfrak{u}$ the projection of $\mathfrak{v}$ onto the line spanned by $\mathfrak{u}$.

### 3.2 Preliminaries and Definitions

Since our methodology is based on techniques in multi-objective minimization (MoM), we first review some concepts from MoM, particularly the multiple gradient descent algorithm (MGDA).

#### 3.2.1 Multi-Objective Minimization for Fairness

Denote by $\boldsymbol{f}(\boldsymbol{\theta}) = \{f_k(\boldsymbol{\theta})\}_{k\in[K]}$ the set of local clients' loss functions; the aim of MoM is to solve

$$\boldsymbol{\theta}^* = \arg\min_{\boldsymbol{\theta}} \boldsymbol{f}(\boldsymbol{\theta}), \tag{1}$$

where the minimization is performed w.r.t. the *partial ordering*. Finding $\boldsymbol{\theta}^*$ could enforce fairness among the users since by setting setting $\boldsymbol{\theta} = \boldsymbol{\theta}^*$, it is not possible to reduce any of the local objective functions $f_k$ without increasing at least another one. Here, $\boldsymbol{\theta}^*$ is called a Pareto-optimal solution of Equation (1). Although finding Pareto-optimal solutions can be challenging, there are several methods to identify the Pareto-stationary solutions instead, which are defined as follows:

**Definition 3.1.** Pareto-stationary (Mukai, 1980): The vector $\boldsymbol{\theta}^*$ is said to be Pareto-stationary iff there exists a convex combination of the gradient-vectors $\{\mathfrak{g}_k(\boldsymbol{\theta}^*)\}_{k\in[K]}$ which is equal to zero; that is, $\sum_{k=1}^K \lambda_k \mathfrak{g}_k(\boldsymbol{\theta}^*) = 0$, where $\boldsymbol{\lambda} \geq 0$, and $\sum_{k=1}^K \lambda_k = 1$.

**Lemma 3.2.** (Mukai, 1980) Any Pareto-optimal solution is Pareto-stationary. On the other hand, if all $\{f_k(\boldsymbol{\theta})\}_{k\in[K]}$'s are convex, then any Pareto-stationary solution is weakly Pareto optimal [2].

There are many methods in the literature to find Pareto-stationary solutions among which MGDA is a popular one (Mukai, 1980; Fliege & Svaiter, 2000; Désidéri, 2012).

---

[2] $\boldsymbol{\theta}^*$ is called a weakly Pareto-optimal solution of Equation (1) if there does not exist any $\boldsymbol{\theta}$ such that $f(\boldsymbol{\theta}) < f(\boldsymbol{\theta}^*)$; meaning that, it is not possible to improve *all* of the objective functions in $f(\boldsymbol{\theta}^*)$. Obviously, any Pareto optimal solution is also weakly Pareto-optimal but the converse may not hold.

MGDA adaptively tunes $\{\lambda_k\}_{k \in [K]}$ by finding the minimal-norm element of the convex hull of the gradient vectors defined as follows (we drop the dependence of $\mathbf{g}_k$ to $\boldsymbol{\theta}^t$ for ease of notation hereafter)

$$\mathcal{G} = \{\boldsymbol{g} \in \mathbb{R}^d | \boldsymbol{g} = \sum_{k=1}^{K} \lambda_k \mathbf{g}_k; \ \lambda_k \geq 0; \ \sum_{k=1}^{K} \lambda_k = 1\}. \tag{2}$$

Denote the minimal-norm element of $\mathcal{G}$ by $\mathfrak{d}(\mathcal{G})$. Then, either (i) $\mathfrak{d}(\mathcal{G}) = 0$, and therefore based on Lemma 3.2 $\mathfrak{d}(\mathcal{G})$ is a Pareto-stationary point; or (ii) $\mathfrak{d}(\mathcal{G}) \neq 0$ and the direction of $-\mathfrak{d}(\mathcal{G})$ is a common descent direction for all the objective functions $\{f_k(\boldsymbol{\theta})\}_{k \in [K]}$ (Désidéri, 2009), meaning that all the directional derivatives $\{\mathbf{g}_k \cdot \mathfrak{d}(\mathcal{G})\}_{k \in [K]}$ are positive. Having positive directional derivatives is a *necessary* condition to ensure that the common direction is descent for all the objective functions.

### 3.2.2 Newton-type methods

First-order FL methods face challenges with slow convergence, measured in terms of the number of iterations or communication rounds. Additionally, their convergence speed is intricately linked to the condition number, influenced by factors such as the model's structure, choice of loss function, and distribution of training data. In contrast, second-order methods exhibit significantly faster performance due to their additional computational effort in estimating the local curvature of the loss landscape. This, in turn, yields faster and more adaptive update directions. Despite requiring more computations per communication round, second-order methods achieve fewer communication rounds. In the context of FL, where communication often poses a bottleneck rather than computation (Yang & Hamidi, 2024; Mohajer Hamidi & Bereyhi, 2024), the appeal of second-order methods has grown. Notably, the Newton's direction is obtained as

$$\mathfrak{d}_N = -(\nabla^2 f(\boldsymbol{\theta}))^{-1} \nabla f(\boldsymbol{\theta}). \tag{3}$$

## 4 Motivation and Methodology

We discuss our motivation in Section 4.1 based on which we elaborate on the inner-working of DQN-FL in Section 4.2.

### 4.1 Motivation

We begin with finding out how much the local loss function $f_k(\cdot)$, $k \in [K]$, changes when the server updates the global model as $\boldsymbol{\theta}^{t+1} = \boldsymbol{\theta}^t - \eta^t \mathfrak{d}^t$ at round $t$. In other words, we want to determine the rate of change $\Delta f_k(\boldsymbol{\theta}^t) \triangleq f_k(\boldsymbol{\theta}^{t+1}) - f_k(\boldsymbol{\theta}^t)$ for the local loss functions. To do this, by writing the first-order Taylor expansion for the local loss function $f_k(\cdot)$, we obtain:

$$f_k(\boldsymbol{\theta}^{t+1}) = f_k(\boldsymbol{\theta}^t - \eta^t \mathfrak{d}^t) \approx f_k(\boldsymbol{\theta}^t) - \eta^t \mathbf{g}_k^t \cdot \mathfrak{d}^t \tag{4}$$
$$\Leftrightarrow \quad \Delta f_k(\boldsymbol{\theta}^t) \approx -\eta^t \mathbf{g}_k^t \cdot \mathfrak{d}^t. \tag{5}$$

As per Equation (5), $f_k(\cdot)$ changes by amount of $-\eta^t \mathbf{g}_k^t \cdot \mathfrak{d}^t$ when the server updates the global model. Hence, if $\mathbf{g}_k^t \cdot \mathfrak{d} \geq 0$, the global updating direction is descent for client $k$, and $\Delta f_k(\boldsymbol{\theta}^t) \leq 0$.

Nevertheless, updating toward a descent direction does not guarantee any meaningful convergence. Indeed, what can guarantee the convergence of GD-like algorithms is the rate of change in the loss function in each iteration[3]. This is in fact what makes the second-order methods to converge faster as the rate of change in the loss functions is automatically determined by the Hessian matrix.

This motivates us to see how the server can update the global model such that the rate of change in the local loss functions is the same as that when local clients update their local loss function using second-order methods.

---

[3]If $f_k(\cdot)$ is $L$-smooth, the convergence of gradient descent algorithm is guaranteed for $\eta^t \in [0, \frac{2}{L}]$.

Specifically, let $d_k^t$ denote the rate of change in local loss function $f_k$ when it updates its local model using Newton method; then, we have

$$d_k^t = \mathbf{g}_k^t \cdot \mathbf{\mathfrak{d}}_N = \mathbf{g}_k^t \cdot \left( (\boldsymbol{H}_k^t)^{-1} \mathbf{g}_k^t \right) = (\mathbf{g}_k^t)^T (\boldsymbol{H}_k^t)^{-1} \mathbf{g}_k^t. \tag{6}$$

Our goal is to assist the server in updating the global model such that, after the update, the rate of change for client $k$ becomes $d_k^t$. Achieving this is not a straightforward task. In the following section, we derive a closed-form solution to meet this criterion.

## 4.2 Methodology

Our method is partially inspired from MGDA algorithm, but incorporates several key modifications. Specifically, our approach comprises two stages: (i) gradient orthogonalization with a tailored scaling strategy; and (ii) finding the optimal weights to combine these orthogonal gradients.

### 4.2.1 *Stage* 1, Gradient Orthogonalization

The clients send the local gradients $\{\mathbf{g}_k\}_{k \in [K]}$ to the server, and then the server first generates a mutually orthogonal [4] set $\{\tilde{\mathbf{g}}_k\}_{k \in [K]}$ that spans the same $K$-dimensional subspace in $\mathbb{R}^d$ as that spanned by $\{\mathbf{g}_k\}_{k \in [K]}$. To this aim, the server exploits a modified Gram–Schmidt orthogonalization process over $\{\mathbf{g}_k\}_{k \in [K]}$ in the following manner [5]

$$\tilde{\mathbf{g}}_1 = \mathbf{g}_1 / d_1^t, \tag{7}$$

$$\tilde{\mathbf{g}}_k = \frac{\mathbf{g}_k - \sum_{i=1}^{k-1} \mathrm{proj}_{\tilde{\mathbf{g}}_i}(\mathbf{g}_k)}{d_k^t - \sum_{i=1}^{k-1} \frac{\mathbf{g}_k \cdot \tilde{\mathbf{g}}_i}{\tilde{\mathbf{g}}_i \cdot \tilde{\mathbf{g}}_i}}, \quad \text{for } k = 2, \ldots, K, \tag{8}$$

where $\gamma > 0$ is a scalar. Note that the orthogonalization approach in *stage* 1 is feasible if we assume that the $K$ gradient vectors $\{\mathbf{g}_k\}_{k \in [K]}$ are linearly independent. Indeed, this assumption is reasonable considering that (i) the gradient vectors $\{\mathbf{g}_k\}_{k \in [K]}$ are $K$ vectors in $d$-dimensional space, and $d >> K$ for the DNNs[6]; and that (ii) the random nature of the gradient vectors due to the non-iid distributions of the local datasets.

### 4.2.2 *Stage* 2, finding optimal weights

In this stage, we aim to find the minimum-norm vector in the convex hull of the *orthogonal* gradients found in *Stage (I)*. First, denote by $\tilde{\mathcal{G}}$ the convex hull of gradient vectors $\{\tilde{\mathbf{g}}_k\}_{k \in [K]}$; that is,

$$\tilde{\mathcal{G}} = \{\boldsymbol{g} \in \mathbb{R}^d | \boldsymbol{g} = \sum_{k=1}^{K} \lambda_k \tilde{\mathbf{g}}_k; \ \lambda_k \geq 0; \ \sum_{k=1}^{K} \lambda_k = 1\}.$$

In the following, we find the minimal-norm element in $\tilde{\mathcal{G}}$, and then we show that this element is a descent direction for all the objective functions.

Denote by $\boldsymbol{\lambda}^*$ the weights corresponding to the minimal-norm vector in $\tilde{\mathcal{G}}$. To find the weight vector $\boldsymbol{\lambda}^*$, we solve

$$\boldsymbol{g}^* = \arg\min_{\boldsymbol{g} \in \mathcal{G}} \|\boldsymbol{g}\|^2, \tag{9}$$

which accordingly finds $\boldsymbol{\lambda}^*$. For an element $\boldsymbol{g} \in \mathcal{G}$, we have

$$\|\boldsymbol{g}\|^2 = \|\sum_{k=1}^{K} \lambda_k \tilde{\mathbf{g}}_k\|^2 = \sum_{k=1}^{K} \lambda_k^2 \|\tilde{\mathbf{g}}_k\|^2, \tag{10}$$

---

[4]Here, orthogonality is in the sense of standard inner product in Euclidean space.

[5]The reason for such normalization will be clarified later.

[6]Also, note that to tackle non-iid distribution of user-specific data, it is a common practice that server selects a different subset of clients in each round (McMahan et al., 2017).

where we used the fact that $\{\tilde{\mathbf{g}}_k\}_{k\in[K]}$ are orthogonal.

To solve Equation (9), we first ignore the inequality $\lambda_k \geq 0$, for $k \in [K]$, and then we observe that it is automatically satisfied. Thus, we make the following Lagrangian to solve the minimization problem in Equation (9):

Hence, $\frac{\partial L}{\partial \lambda_k} = 2\lambda_k \|\tilde{\mathbf{g}}_k\|^2 - \alpha$; and by setting this equation to zero we obtain

$$\lambda_k^* = \frac{\alpha}{2\|\tilde{\mathbf{g}}_k\|^2}. \tag{11}$$

On the other hand, since $\sum_{k=1}^K \lambda_k = 1$, from Equation (11) we have $\alpha = \frac{2}{\sum_{k=1}^K \frac{1}{\|\tilde{\mathbf{g}}_k\|^2}}$ from which the optimal $\boldsymbol{\lambda}^*$ is obtained as follows

$$\lambda_k^* = \frac{1}{\|\tilde{\mathbf{g}}_k\|^2 \sum_{k=1}^K \frac{1}{\|\tilde{\mathbf{g}}_k\|^2}}, \quad \text{for } k \in [K]. \tag{12}$$

Note that $\lambda_k^* > 0$, and therefore the minimum norm vector we found belongs to $\mathcal{G}$. Using the $\boldsymbol{\lambda}^*$ found in (12), we can calculate $\boldsymbol{\mathfrak{d}}^t = \sum_{k=1}^K \lambda_k^* \tilde{\mathbf{g}}_k$ as the minimum norm element in the convex hull $\tilde{\mathcal{G}}$.

**Theorem 4.1.** If the server updates the model toward $\boldsymbol{\mathfrak{d}}^t = \sum_{k=1}^K \lambda_k^* \tilde{\mathbf{g}}_k$, the rate of change for client $k$ is proportional to $d_k^t$, $\forall k \in [K]$.

*Proof.* We shall find the directional derivative of loss function $f_k$, $\forall k \in [K]$, over $\boldsymbol{\mathfrak{d}}^t$:

$$\mathbf{g}_k \cdot \boldsymbol{\mathfrak{d}}^t = \left( \tilde{\mathbf{g}}_k \left( d_k^t - \sum_{i=1}^{k-1} \frac{\mathbf{g}_k \cdot \tilde{\mathbf{g}}_i}{\tilde{\mathbf{g}}_i \cdot \tilde{\mathbf{g}}_i} \right) + \sum_{i=1}^{k-1} \text{proj}_{\tilde{\mathbf{g}}_i}(\mathbf{g}_k) \right) \cdot \left( \sum_{i=1}^K \lambda_i^* \tilde{\mathbf{g}}_i \right) \tag{13}$$

$$= \lambda_k^* \|\tilde{\mathbf{g}}_k\|_2^2 \left( d_k^t - \sum_{i=1}^{k-1} \frac{\mathbf{g}_k \cdot \tilde{\mathbf{g}}_i}{\tilde{\mathbf{g}}_i \cdot \tilde{\mathbf{g}}_i} \right) + \sum_{i=1}^{k-1} \frac{\mathbf{g}_k \cdot \tilde{\mathbf{g}}_i}{\tilde{\mathbf{g}}_i \cdot \tilde{\mathbf{g}}_i} \lambda_i^* \|\tilde{\mathbf{g}}_i\|_2^2 \tag{14}$$

$$= \frac{\alpha}{2} \left( d_k^t - \sum_{i=1}^{k-1} \frac{\mathbf{g}_k \cdot \tilde{\mathbf{g}}_i}{\tilde{\mathbf{g}}_i \cdot \tilde{\mathbf{g}}_i} \right) + \frac{\alpha}{2} \sum_{i=1}^{k-1} \frac{\mathbf{g}_k \cdot \tilde{\mathbf{g}}_i}{\tilde{\mathbf{g}}_i \cdot \tilde{\mathbf{g}}_i} \tag{15}$$

$$= \frac{\alpha}{2} d_k^t = \frac{d_k^t}{\sum_{k=1}^K \frac{1}{\|\tilde{\mathbf{g}}_k\|^2}} > 0, \tag{16}$$

where (i) Equation (13) is obtained by using definition of $\tilde{\mathbf{g}}_k$ in Equation (8), (ii) Equation (14) follows from the orthogonality of $\{\tilde{\mathbf{g}}_k\}_{k=1}^K$ vectors, and (iii) Equation (15) is obtained by using Equation (11). $\square$

Hence, to realize a rate of change similar to the Newton step, at iteration $t$, the server set the global learning rate as $\eta = \sum_{k=1}^K \frac{1}{\|\tilde{\mathbf{g}}_k\|^2}$, and update the global model as:

$$\boldsymbol{\theta}^{t+1} = \boldsymbol{\theta}^t - \eta^t \boldsymbol{\mathfrak{d}}^t = \boldsymbol{\theta}^t - \sum_{k=1}^K \frac{1}{\|\tilde{\mathbf{g}}_k\|^2} \boldsymbol{\mathfrak{d}}^t. \tag{17}$$

> To summarize, updating the global model as in Equation (17) provides two key advantages:
> **(i)** All local loss decreases (as shown by the inequality in Equation (16));
> **(ii)** The rate of change for each local loss function aligns with that of the Newton method.

Similarly to the conventional GD, we note that updating the global model as (17) is a *necessary* condition to have $\boldsymbol{f}(\boldsymbol{\theta}^{t+1}) \leq \boldsymbol{f}(\boldsymbol{\theta}^t)$. In Theorem 4.2 whose proof is differed to Appendix A, we state the *sufficient* condition to satisfy $\boldsymbol{f}(\boldsymbol{\theta}^{t+1}) \leq \boldsymbol{f}(\boldsymbol{\theta}^t)$.

**Theorem 4.2.** Assume that $\boldsymbol{f} = \{f_k\}_{k\in[K]}$ are L-Lipschitz smooth. If the step-size $\eta^t = \sum_{k=1}^K \frac{1}{\|\tilde{\mathbf{g}}_k\|^2} \in [0, \frac{2}{L}\min\{d_k^t\}_{k\in[K]}]$, then $\boldsymbol{f}(\boldsymbol{\theta}^{t+1}) \leq \boldsymbol{f}(\boldsymbol{\theta}^t)$, and equality is achieved iff $\boldsymbol{\mathfrak{d}}^t = \mathbf{0}$.

### 4.3 DQN-Fed Algorithm

Since Newton's method requires the computation of the inverse Hessian matrix, which is computationally expensive, we employ quasi-Newton methods that approximate the inverse of the Hessian using gradient information. The BFGS algorithm (Broyden, 1970) is one such approach. Let $\boldsymbol{B}_k^t$ denote the matrix obtained using BFGS algorithm, where $\boldsymbol{B}_k^t \approx (\boldsymbol{H}_k^t)^{-1}$. Using $\boldsymbol{B}_k^t$, $d_k^t$ in Equation (6) can be approximated by

$$\tilde{d}_k^t = (\mathbf{g}_k^t)^T \boldsymbol{B}_k^t \mathbf{g}_k^t. \tag{18}$$

Lastly, similar to many recent FL algorithms (McMahan et al., 2017; Li et al., 2019a), we allow each client to perform a couple of local epochs $E$ . We summarize DQN-Fed in Algorithm 1.

---

**Algorithm 1:** DQN-Fed.

---

**Input:** Number of global epochs $T$, global learning rate $\eta^t$, number of local epochs $E$, local datasets $\{\mathcal{D}_k\}_{k \in K}$.

**for** $t = 0, 1, \ldots, T - 1$ **do**

    Server randomly selects a subset of devices $\mathcal{S}^t$ and sends $\boldsymbol{\theta}^t$ to them.

    **for** *device $k \in \mathcal{S}^t$ in parallel* **do**

        Set $\hat{\boldsymbol{\theta}}_k^1 = \boldsymbol{\theta}^t$ and $\hat{\boldsymbol{\theta}}_k^0 = \boldsymbol{\theta}^{t-1}$

        **for** $e = 1, 2, \ldots, E$ **do**

            Perform BFGS algorithm as follows

            Set $\mathbf{s}_k^e = \hat{\boldsymbol{\theta}}_k^e - \hat{\boldsymbol{\theta}}_k^{e-1}$, and $\mathbf{y}_k^e = \nabla f(\hat{\boldsymbol{\theta}}_k^e) - \nabla f(\hat{\boldsymbol{\theta}}_k^{e-1})$.

            Iteratively update matrix $\mathbf{B}_k^{e+1}$ using information from $\mathbf{B}_k^e, \mathbf{s}_k^e, \mathbf{y}_k^e$ according to:

$$\mathbf{B}_k^{e+1} = \mathbf{B}_k^e - \frac{\mathbf{B}_k^e \mathbf{s}_k^e (\mathbf{s}_k^e)^\mathsf{T} \mathbf{B}_k^e}{(\mathbf{s}_k^t)^\mathsf{T} \mathbf{B}_k^e \mathbf{s}_k^e} + \frac{\mathbf{y}_k^e (\mathbf{y}_k^e)^\mathsf{T}}{(\mathbf{s}_k^e)^\mathsf{T} \mathbf{y}_k^e}. \tag{19}$$

        **end**

        Use $\mathbf{B}_k^E$ to calculate $\tilde{d}_k^t$ from Equation (18).

        Send local gradient $\mathbf{g}_k = \nabla f(\hat{\boldsymbol{\theta}}_k^E)$ and $\tilde{d}_k^t$ to the server.

    **end**

    Server finds $\{\tilde{\mathfrak{g}}_k\}_{k \in [K]}$ form Equations (7) and (8).

    Server finds $\boldsymbol{\lambda}^*$ from Equation (12).

    Server calculates $\mathfrak{d}^t := \sum_{k=1}^K \lambda_k^* \tilde{\mathfrak{g}}_k$.

    Server updates the global model as $\boldsymbol{\theta}^{t+1} \leftarrow \boldsymbol{\theta}^t - \eta^t \mathfrak{d}^t$.

**end**

**Output:** Global model $\boldsymbol{\theta}^t$.

---

### 4.4 Convergence results

In this section, we analyze the convergence behavior of DQN-Fed by presenting two sets of theorems:

**Set 1:** Theorems 4.3 to 4.5, which focuses on the *fairness* of DQN-Fed and establishes various types of convergence to Pareto-optimal points under different settings. Specifically, we examine the following cases based on how clients update their local models: (i) using SGD with $E = 1$, (ii) using GD with $E > 1$, and (iii) using GD with $E = 1$. Among these, the strongest convergence guarantee is provided for the third scenario.

**Set 2:** Theorem 4.6, which addresses the *optimality gap* by analyzing the number of communication rounds, $T_\epsilon$, required to achieve an $\epsilon$-accurate solution. This provides a convergence rate for DQN-Fed.

**Theorem 4.3** ($E = 1$ & local SGD)**.** Assume that $\boldsymbol{f} = \{f_k\}_{k \in [K]}$ are l-Lipschitz continuous and L-Lipschitz smooth, and that the global step-size $\eta^t$ satisfies the following three conditions: (i) $\eta^t \in (0, \frac{1}{2L}]$, (ii) $\lim_{T \to \infty} \sum_{t=0}^T \eta^t \to \infty$, and (iii) $\lim_{T \to \infty} \sum_{t=0}^T \eta^t \sigma_t < \infty$; where $\sigma_t^2 = \mathbf{E}[\|\tilde{\mathfrak{g}} \boldsymbol{\lambda}^* - \tilde{\mathfrak{g}}_s \boldsymbol{\lambda}_s^*\|]^2$ is the variance of

stochastic common descent direction. Then

$$\lim_{T\to\infty} \min_{t=0,\dots,T} \mathbf{E}[\|\mathfrak{d}^t\|] \to 0. \tag{20}$$

In Theorem 4.3, the variance $\sigma_t^2$ arises from the stochastic nature of the gradient estimates. Controlling $\sigma_t^2$ is intimately related to ensuring that the variance of the stochastic common descent direction diminishes, or at least does not accumulate too quickly. The condition $\lim_{T\to\infty} \sum_{t=0}^{T} \eta^t \sigma_t < \infty$ essentially states that the global step sizes $\eta^t$ must shrink at a rate that sufficiently "smooths out" variance over time.

But how can the condition $\lim_{T\to\infty} \sum_{t=0}^{T} \eta^t \sigma_t < \infty$ be guaranteed in practice? In the following, we provide two possible ways to meet this condition:

First, by choosing a non-increasing global step-size schedule $\{\eta^t\}$ that decays to zero at a sufficiently fast rate, the effective "weighted sum" $\sum_{t=0}^{T} \eta^t \sigma_t < \infty$ remains finite. For example, if $\eta^t = \mathcal{O}(t^{-\alpha})$, for some $\alpha > 0$, and if $\sigma_t$ does not grow faster than $\mathcal{O}(t^\beta)$ for some $\beta < \alpha$, then $\sum_t \eta^t \sigma_t$ will converge.

Second, implementing variance reduction methods (e.g., control variates, variance-reduced stochastic gradient estimators) within local updates can ensure that $\sigma_t$ does not persistently remain large. In federated settings, this can be achieved by techniques such as periodically synchronizing with a global model (which reduces drift and variance), incorporating momentum-based methods, or employing mini-batching strategies that limit stochastic fluctuations.

**Theorem 4.4** ($E > 1$ & local GD)**.** Assume that $\boldsymbol{f} = \{f_k\}_{k\in[K]}$ are l-Lipschitz continuous and L-Lipschitz smooth. Denote by $\eta^t$ and $\eta$ the global and local learning rates, respectively. Also, define $\zeta^t = \|\boldsymbol{\lambda}^* - \boldsymbol{\lambda}_E^*\|$, where $\boldsymbol{\lambda}_E^*$ is the optimum weights obtained from pseudo-gradients after $E$ local epochs. We have

$$\lim_{T\to\infty} \min_{t=0,\dots,T} \|\mathfrak{d}^t\| \to 0, \tag{21}$$

if the following conditions are satisfied: (i) $\eta^t \in (0, \frac{1}{2L}]$, (ii) $\lim_{T\to\infty} \sum_{t=0}^{T} \eta^t \to \infty$, (iii) $\lim_{t\to\infty} \eta^t \to 0$, (iv) $\lim_{t\to\infty} \eta \to 0$, and (v) $\lim_{t\to\infty} \zeta^t \to 0$.

In Theorem 4.4, $\zeta^t = \|\boldsymbol{\lambda}^* - \boldsymbol{\lambda}_E^*\|$ measures the deviation between the true optimum $\boldsymbol{\lambda}^*$ and the optimum obtained from the pseudo-gradients after $E$ local epochs, $\boldsymbol{\lambda}_E^*$. The condition $\lim_{t\to\infty} \zeta^t \to 0$ reflects the requirement that local model updates become increasingly aligned with the true global optimum as training progresses. Below, we discuss two strategies to ensure this condition is satisfied.

First, as stated, both the global and local learning rates should diminish over time: $\lim_{t\to\infty} \eta^t \to 0$ and $\lim_{t\to\infty} \eta \to 0$. As these rates decrease, the updates become more conservative, allowing the local solutions $\boldsymbol{\lambda}_E^*$ to approach the global solution $\boldsymbol{\lambda}^*$.

Second, adjusting the frequency of global synchronization over time helps ensure that local models do not drift too far from the global model. By increasing synchronization frequency or using adaptive synchronization strategies as training progresses, one can reduce the gap $\zeta^t$.

Before introducing Theorem 4.5, we first introduce some notations. Denote by $\vartheta$ the Pareto-stationary solution set[7] of minimization problem $\arg\min_{\boldsymbol{\theta}} \boldsymbol{f}(\boldsymbol{\theta})$. Then, denote by $\boldsymbol{\theta}^*$ the projection of $\boldsymbol{\theta}^t$ onto the set $\vartheta$; that is, $\boldsymbol{\theta}^* = \arg\min_{\boldsymbol{\theta}\in\vartheta} \|\boldsymbol{\theta}^t - \boldsymbol{\theta}\|^2$.

**Theorem 4.5** ($E = 1$ & local GD)**.** Assume that $\boldsymbol{f} = \{f_k\}_{k\in[K]}$ are l-Lipschitz continuous and $\sigma$-convex, and that the global step-size $\eta^t$ satisfies the following two conditions: (i) $\lim_{t\to\infty} \sum_{j=0}^{t} \eta_j \to \infty$, and (ii) $\lim_{t\to\infty} \sum_{j=0}^{t} \eta_j^2 < \infty$. Then almost surely $\boldsymbol{\theta}^t \to \boldsymbol{\theta}^*$; that is,

$$\mathbb{P}\left(\lim_{t\to\infty} (\boldsymbol{\theta}^t - \boldsymbol{\theta}^*) = 0\right) = 1, \tag{22}$$

where $\mathbb{P}(\mathcal{E})$ denotes the probability of event $\mathcal{E}$.

---

[7]In general, the Pareto-stationary solution of multi-objective minimization problem forms a set with cardinality of infinity (Mukai, 1980).

The proofs for Theorems 4.3 to 4.5 are provided in Appendices B.1 to B.3, respectively. Note that all the Theorems 4.3 to 4.5 provide some types of convergence to a Pareto-optimal solution of optimization problem in Equation (1). Specifically, diminishing $\mathfrak{d}^t$ in Theorems 4.3 and 4.4 implies that we are reaching to a Pareto-optimal point (Désidéri, 2009). On the other hand, Theorem 4.5 explicitly provides this convergence guarantee in an almost surely fashion.

In addition, the following theorem shows that DQN-Fed has a *linear-quadratic* convergence rate.

**Theorem 4.6** (Convergence **rate** of DQN-Fed)**.** Assume that $E = 1$ and clients perform local GD. In addition, assume that the global loss function is twice continuously differentiable, $L$-Lipschitz gradient ($L$-smooth) and $\lambda$-strongly convex. Note that the strong convexity of the global loss function implies that there exists a unique optimal model parameter, which we denote by $\boldsymbol{\theta}^{\mathsf{opt}}$. In addition, assume that the matrix $\mathbf{B}_t^{-1}$ is a $\delta$-approximate of true inverse Hessian $\mathbf{H}_t^{-1}$; that is $\|\mathbf{B}_t^{-1} - \mathbf{H}_t^{-1}\| \leq \delta\|\mathbf{H}_t^{-1}\|$. Then,

$$\|\boldsymbol{\theta}_t - \boldsymbol{\theta}^{\mathsf{opt}}\| \leq \begin{cases} \left(\frac{L\delta}{\lambda}\right)^t \|\boldsymbol{\theta}_0 - \boldsymbol{\theta}^{\mathsf{opt}}\| + A_0', & t \leq t_0 \\ \left(\frac{L\delta}{\lambda}\right)^t \|\boldsymbol{\theta}_0 - \boldsymbol{\theta}^{\mathsf{opt}}\| + A_1', & t > t_0 \end{cases} \tag{23}$$

where $A_0'$ and $A_1'$ are defined as follows

$$A_0' = \frac{\left(\frac{L\delta}{\lambda}\right)^t - 1}{\frac{L\delta}{\lambda} - 1}\left[\frac{\lambda}{L}(t_0 - t + \frac{2\gamma}{1 - \gamma})\right], \tag{24a}$$

$$A_1' = \frac{\left(\frac{L\delta}{\lambda}\right)^t - 1}{\frac{L\delta}{\lambda} - 1}\left[\frac{2\lambda\gamma^{2^{t-t_0}}}{L(1 - \gamma^{2^{t-t_0}})} + \right], \tag{24b}$$

$$\text{with} \qquad t_0 = \max\left\{0, \lceil\frac{2L}{\lambda^2\|\mathfrak{d}_0\|}\rceil - 2\right\}, \tag{24c}$$

$$\text{and} \qquad \gamma = \frac{L}{2\lambda^2}\|\mathfrak{d}_0\| - \frac{t_0}{4}. \tag{24d}$$

*Proof.* The proof is differed to Appendix D. $\qquad\square$

As per Theorem (4.6), DQN-Fed method has a linear-quadratic convergence rate. In fact, the quadratic term in (23) is exactly the same as that of Polyak & Tremba (2020); yet, the linear term is the result of approximating the local Hessian matrices using BFGS method.

In the following corollaries, our objective is to determine the required number of communication rounds $T_\epsilon$ such that $\|\boldsymbol{\theta}_{T_\epsilon} - \boldsymbol{\theta}^{\mathsf{opt}}\| \leq \epsilon$.

**Corollary 4.7.** [*Quadratic convergence rate*] If $\|\boldsymbol{\theta}_0 - \boldsymbol{\theta}^{\mathsf{opt}}\| < \frac{A_1'}{\left(\frac{L\delta}{\lambda}\right)^t}$, then DQN-Fed has a quadratic convergence rate:

$$\|\boldsymbol{\theta}_t - \boldsymbol{\theta}^{\mathsf{opt}}\| \leq 2A_1'. \tag{25}$$

Also, if $\frac{L\delta}{\lambda} < 1$, we have

$$T_\epsilon = \mathcal{O}\left(\log\log\frac{1}{\epsilon}\right), \tag{26}$$

which is also called super-linear convergence rate.

**Corollary 4.8.** [*Linear convergence rate*] On the other hand, If $\|\boldsymbol{\theta}_0 - \boldsymbol{\theta}^{\mathsf{opt}}\| \geq \frac{A_1'}{\left(\frac{L\delta}{\lambda}\right)^t}$ and $\frac{L\delta}{\lambda} < 1$, then DQN-Fed method has a linear convergence rate:

$$\|\boldsymbol{\theta}_t - \boldsymbol{\theta}^{\mathsf{opt}}\| \leq 2\left(\frac{L\delta}{\lambda}\right)^t \|\boldsymbol{\theta}_0 - \boldsymbol{\theta}^{\mathsf{opt}}\|. \tag{27}$$

and,

$$T_\epsilon = \mathcal{O}\left(\frac{1}{\log(\frac{\lambda}{L\delta})}\log\frac{1}{\epsilon}\right). \tag{28}$$

The proof for Corollary 4.7 and 4.8 can be found in Appendix E and F, respectively.

**Remark 4.9.** It is worth noting that for distributed GD-like methods, the number of communication rounds needed to achieve a desired precision $\epsilon$, follows a linear convergence rate. Specifically, we have $T_\epsilon = \mathcal{O}\left(\frac{L}{\lambda} \log \frac{1}{\epsilon}\right)$. This underscores the superiority of DQN-Fed in terms of convergence rate.

## 5 Experiments

In this section, we conclude the paper by presenting a series of experiments to demonstrate the performance of DQN-Fed. We also conduct a comparative analysis to assess its effectiveness against state-of-the-art alternatives using various performance metrics.

• **Datasets:** We conduct a comprehensive set of experiments across **seven** datasets. In this section, we present results for four datasets: CIFAR-$\{10, 100\}$ (Krizhevsky et al., 2009), FEMNIST (Caldas et al., 2018), and Shakespeare (McMahan et al., 2017). Results for Fashion MNIST (Xiao et al., 2017), TinyImageNet (Le & Yang, 2015), and CINIC-10 (Darlow et al., 2018) are discussed in Appendix G. To demonstrate DQN-Fed's effectiveness across different FL scenarios, we examine two FL setups for each dataset in this section. Furthermore, we evaluate DQN-Fed's performance on a real-world noisy dataset, Clothing1M (Xiao et al., 2015), in Appendix I.

• **Benchmarks:** We compare the performance of DQN-Fed against some fair first-order FL and some second-order FL methods. The fair FL algorithms include q-FFL (Li et al., 2019a), TERM (Li et al., 2020), FedMGDA+ (Hu et al., 2022), Ditto (Li et al., 2021), FedLF (Pan et al., 2024), FedHEAL (Chen et al., 2024), and conventional FedAvg (McMahan et al., 2017); and also second-order FL methods include FedNL (Safaryan et al., 2022) and FedNew (Elgabli et al., 2022).

It is worth noting that we conduct a grid-search to find the best hyper-parameters for each of the benchmark methods including DQN-Fed in our experiments. The details of this hyper-parameter tuning are reported in Appendix H.

• **Performance metrics:** Denote by $a_k$ the prediction accuracy on device $k$. We use $\bar{a} = \frac{1}{K} \sum_{k=1}^{K} a_k$ as the average test accuracy of the underlying FL algorithm, and use $\sigma_a = \sqrt{\frac{1}{K} \sum_{k=1}^{K} (a_k - \bar{a})^2}$ as the standard deviation of the accuracy across the clients. Furthermore, we report Worst 10% (5%) and Best 10% (5%) accuracies as a common metric in fair FL algorithms (Li et al., 2020).

• **Notations:** We use **bold** and underlined numbers to denote the best and second best performance, respectively. We use $e$ and $K$ to represent the number of local epochs and that of clients, respectively.

### 5.1 CIFAR-10

CIFAR-10 dataset (Krizhevsky et al., 2009) has 50K training and 10K test images of size $32 \times 32$ labeled for 10 classes. The batch size is equal to 64 for both of the following setups.

• **Setup 1:** Following Wang et al. (2021b), we sort the dataset based on their classes, and then split them into 200 shards. Each client randomly selects two shards without replacement so that each has the same local dataset size. We use a feedforward neural network with 2 hidden layers. We fix $E = 1$ and $K = 100$. We carry out 2000 rounds of communication, and sample 10% of the clients in each round. We run SGD on local datasets with stepsize $\eta = 0.1$.

• **Setup 2:** We distribute the dataset among the clients deploying Dirichlet allocation (Wang et al., 2020) with $\beta = 0.5$. We use ResNet-18 (He et al., 2016) with Group Normalization (Wu & He, 2018). We perform 100 communication rounds in each of which all clients participate. We set $E = 1$, $K = 10$ and $\eta = 0.01$.

### 5.1.1 Wall-Clock Time

For the first setting, we also measure and report the wall-clock time (in seconds) for both the benchmark methods and DQN-Fed. These results, presented in Table 2, were obtained using a single NVIDIA RTX 3090

Table 1: Test accuracy on CIFAR-10. The reported results are averaged over 5 seeds.

| | | Setup 1 | | | | Setup 2 | | | |
|---|---|---|---|---|---|---|---|---|---|
| | Algorithm | $\bar{a}$ | $\sigma_a$ | W(5%) | B(5%) | $\bar{a}$ | $\sigma_a$ | W(10%) | B(10%) |
| Naive First-Order | FedAvg | 46.85 | 3.54 | 19.84 | 69.28 | 63.55 | 5.44 | 53.40 | 72.24 |
| Fair First-Order | q-FFL | 46.30 | 3.27 | 23.39 | 68.02 | 57.27 | 5.60 | 47.29 | 66.92 |
| | FedMGDA | 45.34 | 3.37 | 24.00 | 68.51 | 62.05 | **4.88** | 52.69 | 70.77 |
| | FedHEAL | 46.40 | 3.61 | 19.33 | 69.30 | 63.05 | 4.95 | 48.69 | 70.88 |
| | TERM | 47.11 | 3.66 | 28.21 | 69.51 | 64.15 | 5.90 | 56.21 | 72.20 |
| | Ditto | 46.31 | 3.44 | 27.14 | 68.44 | 63.49 | 5.70 | 55.99 | 71.34 |
| Second-Order | FedNL | 47.33 | 3.92 | 24.41 | 69.52 | 64.72 | 6.02 | 56.20 | 72.33 |
| | FedNew | 47.51 | 3.68 | 25.77 | **69.74** | 64.58 | 6.11 | 56.96 | 72.12 |
| | DQN-Fed | **47.72** | **3.20** | **29.34** | 69.37 | **64.88** | 4.90 | **58.01** | **72.88** |

GPU. As shown, the second-order methods (last three columns) exhibit higher computational times. Notably, among the second-order methods, DQN-Fed demonstrates the fastest runtime.

Table 2: Wall-clock time (in seconds) for benchmark methods and DQN-Fed on CIFAR-10, Setup 1.

| Algorithm | FedAvg | q-FFL | FedMGDA | TERM | Ditto | FedNL | FedNew | DQN-Fed |
|---|---|---|---|---|---|---|---|---|
| Wall-clock (s) | 1250 | 1360 | 1333 | 1312 | 1320 | 2320 | 2140 | 1810 |

### 5.1.2 Ablation Study on the Percentage of Client Participation

In this subsection, we examine the impact of varying client participation rates on the performance of DQN-Fed and the benchmark methods. Specifically, we reproduce the results from CIFAR-10 (Setup 2) using 10% and 50% client participation, while the previously reported results in Table 1 correspond to 100% participation. The results are summarized in Table 3.

The following observations are made:

(i) DQN-Fed continues to outperform the benchmark methods in both fairness and average accuracy across all participation levels.

(ii) As the percentage of client participation decreases, the performance of all methods shows a slight decline, highlighting the importance of client participation in federated learning settings.

Table 3: Ablation study on client participation for Setup 2 on CIFAR-10. Test accuracy results are averaged over 5 seeds.

| Participation | Algorithm | 10% Participation | | | 50% Participation | | |
|---|---|---|---|---|---|---|---|
| | | $\bar{a}$ | $\sigma_a$ | B(10%) | $\bar{a}$ | $\sigma_a$ | B(10%) |
| **Naive First-Order** | FedAvg | 61.41 | 6.36 | 68.12 | 62.74 | 6.01 | 70.15 |
| **Fair First-Order** | q-FFL | 56.55 | 6.24 | 65.89 | 58.97 | 5.95 | 68.22 |
| | FedMGDA | 61.05 | 5.80 | 69.07 | 62.57 | 5.54 | 70.52 |
| | FedHEAL | 61.26 | 6.12 | 68.52 | 62.92 | 5.71 | 70.72 |
| | TERM | 63.22 | 5.93 | 70.56 | 63.81 | 5.66 | 71.83 |
| | Ditto | 62.87 | 6.07 | 69.92 | 63.51 | 5.76 | 71.24 |
| **Second-Order** | FedNL | 63.25 | 6.16 | 70.81 | 64.15 | 5.90 | 71.92 |
| | FedNew | 63.80 | 6.07 | 71.21 | 64.56 | 5.80 | 72.08 |
| | DQN-Fed | **64.14** | **5.67** | **72.21** | **64.71** | **5.48** | **72.55** |

## 5.2 CIFAR-100

CIFAR-100 (Krizhevsky et al., 2009) has the same number of samples as CIFAR-10, but comprises 100 classes compared to the 10 classes found in CIFAR-10.

The model for both setups is ResNet-18 (He et al., 2016) with Group Normalization (Wu & He, 2018), where all clients participate in each round. We also set $E = 1$ and $\eta = 0.01$. The batch size is equal to 64. The results are reported in Table 4 for both of the following setups:

• **Setup 1**: We set $K = 10$ and $\beta = 0.5$ for Dirichlet allocation, and use 400 communication rounds.

• **Setup 2**: We set $K = 50$ and $\beta = 0.05$ for Dirichlet allocation, and use 200 communication rounds.

Table 4: Test accuracy on CIFAR-100. The reported results are averaged over 5 different seeds.

| | | Setup 1 | | | | Setup 2 | | | |
|---|---|---|---|---|---|---|---|---|---|
| | Algorithm | $\bar{a}$ | $\sigma_a$ | W(10%) | B(10%) | $\bar{a}$ | $\sigma_a$ | W(10%) | B(10%) |
| Naive First-Order | FedAvg | 30.05 | 4.03 | 25.20 | 40.31 | 20.15 | 6.40 | 11.20 | 33.80 |
| Fair First-Order | q-FFL | 28.86 | 4.44 | 25.38 | 39.77 | 20.20 | 6.24 | 11.09 | 34.02 |
| | FedMGDA | 29.12 | 4.17 | 25.67 | 39.71 | 20.15 | 5.41 | 11.12 | 33.92 |
| | FedLF | 30.28 | 3.68 | 25.33 | 39.45 | 18.92 | 4.90 | 11.29 | 28.60 |
| | TERM | 30.34 | **3.51** | 27.03 | 39.35 | 17.88 | 5.98 | 10.09 | 31.68 |
| | Ditto | 29.81 | 3.79 | 26.90 | 39.39 | 17.52 | 5.65 | 10.21 | 31.25 |
| Fair First-Order | FedNL | 31.58 | 4.55 | 27.14 | 40.62 | 22.74 | 6.02 | 12.15 | 34.44 |
| | FedNew | 30.95 | 4.39 | 27.19 | 40.55 | 21.16 | 5.27 | 11.77 | 34.27 |
| | DQN-Fed | **32.58** | 3.60 | **27.91** | **40.99** | **23.15** | **4.45** | **12.81** | **35.11** |

## 5.3 FEMNIST

FEMNIST (Federated Extended MNIST) (Caldas et al., 2018) is a federated image dataset distributed over 3,550 devices which has 62 classes containing $28 \times 28$-pixel images of digits (0-9) and English characters (A-Z, a-z). For implementation, we use a CNN model with 2 convolutional layers followed by 2 fully-connected layers. The batch size is 32, and $E = 2$ for both of the following setups:

• **FEMNIST-original:** We use the setting in Li et al. (2021), and randomly sample $K = 500$ devices and train models using the default data stored in each device.

• **FEMNIST-skewed:** $K = 100$. We sample 10 lower case characters ('a'-'j') from Extended MNIST (EMNIST), and randomly assign 5 classes to each of the 100 devices.

Consistent with Li et al. (2019a), we use two other fairness metrics for this dataset: (i) the angle between the accuracy distribution and the all-ones vector **1** denoted by Angle (°), and (ii) the KL divergence between the normalized accuracy $a$ and uniform distribution $u$ denoted by KL $(a\|u)$. Results for both setups are reported in Table 5.

## 5.4 Text Data

We use *The Complete Works of William Shakespeare* (McMahan et al., 2017) as the dataset, and train an RNN whose input is 80-character sequence to predict the next character. We use $E = 1$, and let all the devices participate in each round. The results are reported in Table 6 for the following setups:

• **Setup 1**: Following McMahan et al. (2017), we subsample 31 speaking roles, and assign each role to a client ($K = 31$) to complete 500 communication rounds. We use a model with two LSTM layers (Hochreiter & Schmidhuber, 1997) and one densely-connected layer. The initial $\eta = 0.8$ with decay rate of 0.95.

Table 5: Test accuracy on FEMNIST. The reported results are averaged over 5 different seeds.

| | | FEMNIST-**original** | | | | FEMNIST-**skewed** | | | |
|---|---|---|---|---|---|---|---|---|---|
| | Algorithm | $\bar{a}$ | $\sigma_a$ | Ang (°) | KL $(a\|u)$ | $\bar{a}$ | $\sigma_a$ | Ang (°) | KL $(a\|u)$ |
| Naive First-Order | FedAvg | 80.42 | 11.16 | 10.18 | 0.017 | 79.24 | 22.30 | 12.29 | 0.054 |
| Fair First-Order | q-FFL | 80.91 | 10.62 | 9.71 | 0.016 | 84.65 | 18.56 | 12.01 | 0.038 |
| | FedMGDA | 81.00 | 10.41 | 10.04 | 0.016 | 85.41 | 17.36 | 11.63 | 0.032 |
| | TERM | 81.08 | 10.32 | 9.15 | 0.015 | 84.29 | **13.88** | **11.27** | 0.025 |
| | FedLF | 82.45 | 9.85 | 9.01 | 0.012 | 85.21 | 14.92 | 11.44 | 0.027 |
| | Ditto | 83.77 | 10.13 | 9.34 | 0.014 | 92.51 | 14.32 | 11.45 | 0.022 |
| Fair First-Order | FedNL | 84.21 | 11.22 | 10.07 | 0.015 | 92.94 | 16.45 | 12.56 | 0.045 |
| | FedNew | 84.25 | 10.88 | 9.78 | 0.014 | 92.25 | 15.21 | 11.92 | 0.037 |
| | DQN-Fed | **85.15** | **9.58** | **8.14** | **0.010** | **93.80** | 13.91 | 11.41 | **0.011** |

• **Setup 2**: Among the 31 speaking roles, the 20 ones with more than 10000 samples are selected, and assigned to 20 clients. We use an LSTM followed by a fully-connected layer. $\eta = 2$, and the number of communication is 100.

Table 6: Test accuracy on Shakespeare. The reported results are averaged over 5 different seeds.

| | | Setup 1 | | | | Setup 2 | | | |
|---|---|---|---|---|---|---|---|---|---|
| | Algorithm | $\bar{a}$ | $\sigma_a$ | W(10%) | B(10%) | $\bar{a}$ | $\sigma_a$ | W(10%) | B(10%) |
| Naive First-Order | FedAvg | 53.21 | 9.25 | 51.01 | 58.41 | 50.48 | 1.24 | 48.20 | 52.10 |
| Fair First-Order | q-FFL | 53.90 | 7.52 | 51.52 | 58.47 | 50.72 | 1.07 | 48.90 | 52.29 |
| | FedMGDA | 53.08 | 8.14 | 52.84 | 58.51 | 50.41 | 1.09 | 48.18 | 51.99 |
| | FedLF | 54.58 | 8.44 | 52.87 | 59.84 | 52.45 | 1.23 | 50.02 | 54.17 |
| | TERM | 54.16 | 8.21 | 52.09 | 59.15 | 52.17 | 1.11 | 49.14 | 53.62 |
| | Ditto | 60.74 | 8.32 | 53.57 | **64.92** | **53.12** | 1.20 | 50.94 | **55.23** |
| Fair First-Order | FedNL | 60.25 | 8.24 | 53.15 | 64.15 | 52.24 | 1.25 | 50.77 | 54.41 |
| | FedNew | 60.59 | 7.55 | 53.18 | 64.09 | 52.49 | 1.19 | 50.82 | 54.36 |
| | DQN-Fed | **61.65** | **6.55** | **53.79** | 64.86 | 52.89 | **0.98** | **51.02** | 54.48 |

# 6 Experimental Analysis

## 6.1 Analysis of Results

Based on the insights gleaned from Tables 1 and 4 to 6, several noteworthy observations emerge:

(*i*) Naive second-order FL methods, namely FedNL and FedNew, tend to train unfair models, despite achieving high average accuracy across clients.

(*ii*) Compared to the benchmark models, DQN-Fed consistently trains models that demonstrate significantly higher levels of fairness across clients.

(*iii*) The average accuracy of the model learned by DQN-Fed is higher compared to both first-order and second-order FL methods.

## 6.2 Comparison of Convergence Rate

In this subsection, we empirically compare the convergence speed of DQN-Fed against several fair first-order methods. To do so, we use the four datasets from setup 1 described in Section 5 and plot the validation accuracy of different methods as a function of communication rounds. The results are shown in Figure 1. As observed, DQN-Fed demonstrates a faster convergence rate compared to all benchmark methods.

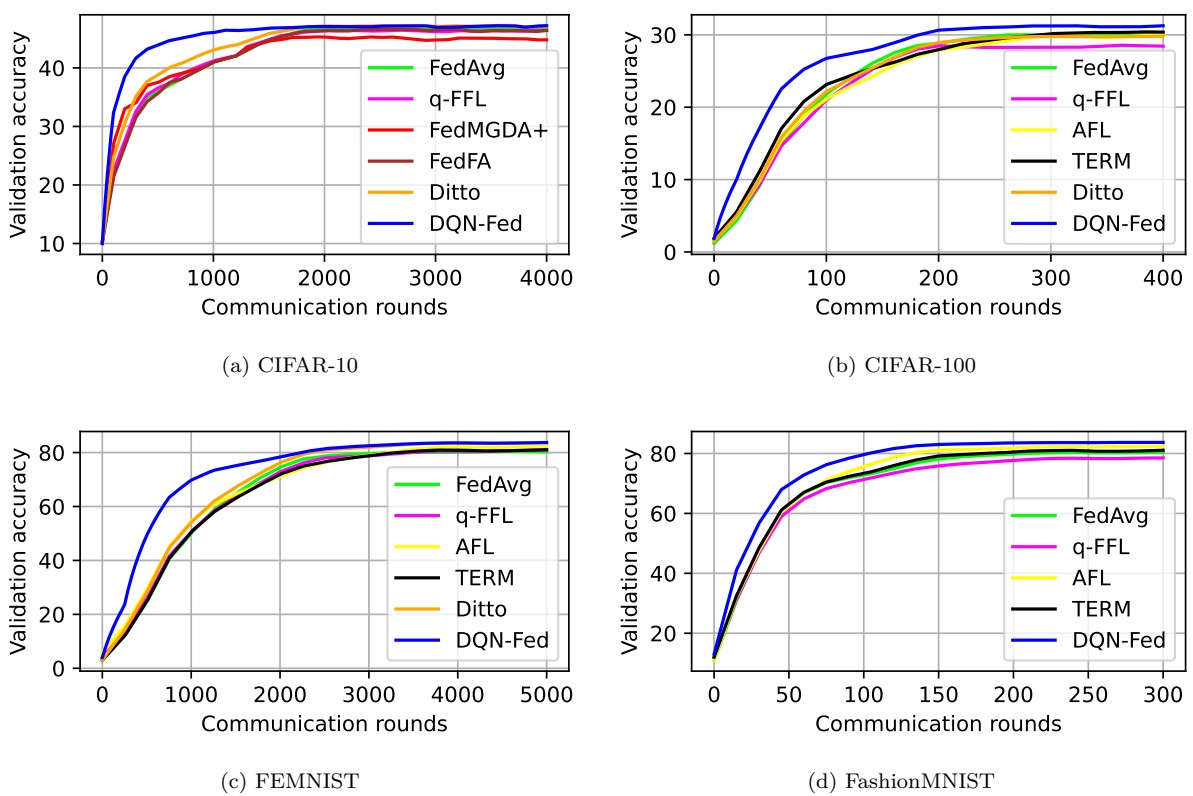

(a) CIFAR-10

(b) CIFAR-100

(c) FEMNIST

(d) FashionMNIST

Figure 1: The test accuracy curves Vs. communication rounds for DQN-Fed and the benchmark methods.

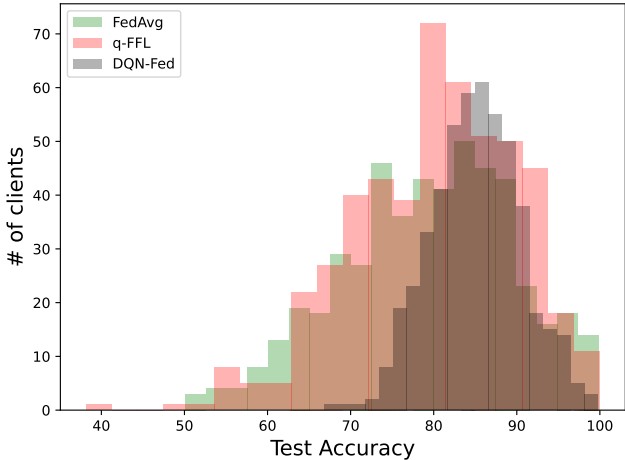

Figure 2: The histogram of clients accuracy for models trained via FedAvg, q-FFL and DQN-Fed.

## 6.3   Histogram of the Clients Accuracy

To gain deeper insight into the variability in client accuracy, we plot the accuracy histograms of 500 clients drawn from the original FEMNIST dataset. In particular, we compare three methods—(i) FedAvg, (ii) q-FFL, and (iii) DQN-Fed—each optimized with well-tuned hyperparameters. As shown in Figure 2, the accuracy distribution under DQN-Fed is notably more concentrated, indicating a fairer spread of performance across clients.

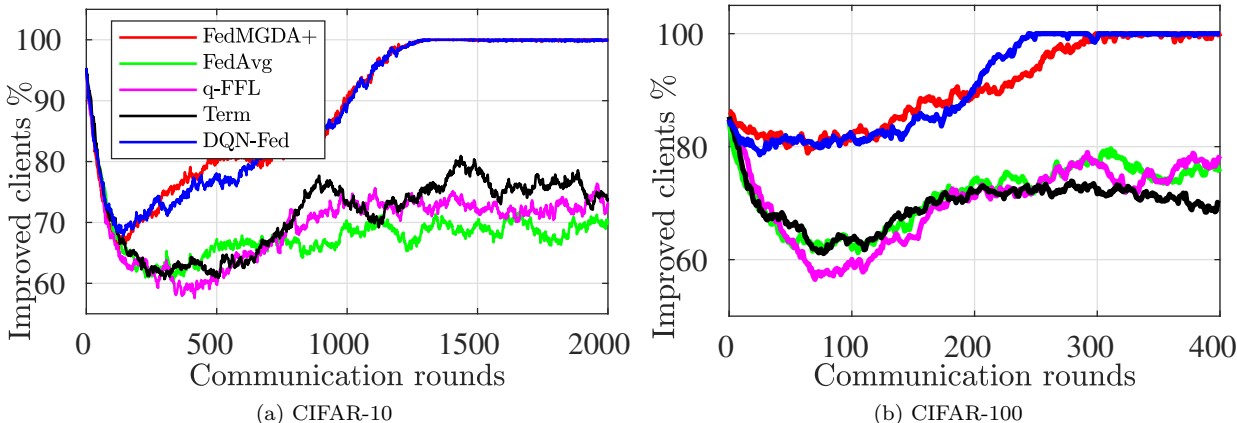

Figure 3: The number of improved clients Vs. communication rounds for DQN-Fed and the benchmark methods.

### 6.4 Percentage of Improved Clients

We measure the training loss before and after each communication round for all participating clients and report the percentage of clients whose loss function decreased or remained unchanged, defined as

$$\rho_t = \frac{\sum_{k \in \mathcal{S}_t} \mathbb{I}\{\boldsymbol{f}_k(\boldsymbol{\theta}^{t+1}) \leq \boldsymbol{f}_k(\boldsymbol{\theta}^t)\}}{|\mathcal{S}_t|}, \tag{29}$$

where $\mathcal{S}_t$ is the participating clients in round $t$, and $\mathbb{I}(\cdot)$ is the indicator function. Then, we plot $\rho_t$ versus communication rounds for different fair FL methods. The curves for CIFAR-10 and CIFAR-100 datasets are reported in Figure 3a and Figure 3b, respectively. As seen, both DQN-Fed and FedMGDA+ consistently outperform other benchmark methods in that fewer clients' performances get worse after participation. We further note that after enough number of communication rounds, curves for both DQN-Fed and FedMGDA+ converge to 100%.

## 7 Conclusion

This paper introduced distributed quasi-Newton federated learning (DQN-Fed), a novel approach designed to ensure fairness while harnessing the fast convergence properties of quasi-Newton methods in FL setting. DQN-Fed aids the server in updating the global model by ensuring (i) all local loss functions decrease, promoting fairness; and (ii) the rate of change in local loss functions matches that of the quasi-Newton method. We proved the convergence of DQN-Fed and establish its linear-quadratic convergence rate. Furthermore, we validated DQN-Fed's effectiveness across various federated datasets, demonstrating its superiority over state-of-the-art fair FL methods.

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

# A   Proof of Theorem 4.2

*Proof.* If all the $\{f_k\}_{k\in[K]}$ are $L$-smooth, then

$$\boldsymbol{f}(\boldsymbol{\theta}^{t+1}) \leq \boldsymbol{f}(\boldsymbol{\theta}^t) + \mathfrak{g}^T(\boldsymbol{\theta}^{t+1} - \boldsymbol{\theta}^t) + \frac{L}{2}\|\boldsymbol{\theta}^{t+1} - \boldsymbol{\theta}^t\|^2. \tag{30}$$

Now, for client $k \in [K]$, by using the update rule Equation (17) in Equation (30) we obtain

$$f_k(\boldsymbol{\theta}^{t+1}) \leq f_k(\boldsymbol{\theta}^t) - \eta^t \mathfrak{g}_k \cdot \mathfrak{d}^t + (\eta^t)^2 \frac{L}{2}\|\mathfrak{d}^t\|^2. \tag{31}$$

To impose $f_k(\boldsymbol{\theta}^{t+1}) \leq f_k(\boldsymbol{\theta}^t)$, we should have

$$\eta^t \mathfrak{g}_k \cdot \mathfrak{d}^t \geq (\eta^t)^2 \frac{L}{2}\|\mathfrak{d}^t\|^2 \tag{32}$$

$$\Leftrightarrow \quad \mathfrak{g}_k \cdot \mathfrak{d}^t \geq \frac{\eta^t L}{2} \sum_{k=1}^{K} \frac{\|\tilde{\mathfrak{g}}_k\|^2}{\|\tilde{\mathfrak{g}}_k\|^4 \left(\sum_{i=1}^{K} \frac{1}{\|\tilde{\mathfrak{g}}_i\|^2}\right)^2} \tag{33}$$

$$\Leftrightarrow \quad \frac{\tilde{d}_k^t}{\sum_{k=1}^{K} \frac{1}{\|\tilde{\mathfrak{g}}_k\|^2}} \geq \frac{\eta^t L}{2} \frac{1}{\left(\sum_{k=1}^{K} \frac{1}{\|\tilde{\mathfrak{g}}_k\|^2}\right)^2} \sum_{k=1}^{K} \frac{1}{\|\tilde{\mathfrak{g}}_k\|^2} \tag{34}$$

$$\Leftrightarrow \quad \eta^t \leq \frac{2}{L}\tilde{d}_k^t. \tag{35}$$

$\square$

# B   Convergence of DQN-Fed

In the following, we provide three theorems to analyse the convergence of DQN-Fed under different scenarios. Specifically, we consider three cases: (i) Theorem B.1 considers $E = 1$ and using SGD for local updates, (ii) Theorem B.2 considers an arbitrary value for $e$ and using GD for local updates, and (iii) Theorem B.4 considers $E = 1$ and using GD for local updates.

## B.1   Case 1: $E = 1$ & local SGD

**Notations:** We use subscript $(\cdot)_s$ to indicate a stochastic value. Using this notation for the values we introduced in the paper, our notations used in the proof of Theorem B.1 are summarized in Table 7.

Table 7: Notations used in Theorem B.1 for $E = 1$ & local SGD.

| Notation | Description |
|---|---|
| $\mathfrak{g}_{k,s}$ | **Stochastic** gradient vector of client $k$. |
| $\mathfrak{g}_s$ | Matrix of **Stochastic** gradient vectors $[\mathfrak{g}_{1,s}, \ldots, \mathfrak{g}_{K,s}]$. |
| $\tilde{\mathfrak{g}}_{k,s}$ | **Stochastic** gradient vector of client $k$ after orthogonalization process. |
| $\tilde{\mathfrak{g}}_s$ | Matrix of orthogonalized **Stochastic** gradient vectors $[\tilde{\mathfrak{g}}_{1,s}, \ldots, \tilde{\mathfrak{g}}_{K,s}]$. |
| $\lambda_{k,s}^*$ | Optimum weights obtained from Equation (12) using **Stochastic** gradients $\tilde{\mathfrak{g}}_s$. |
| $\mathfrak{d}_s$ | Optimum direction obtained using **Stochastic** $\tilde{\mathfrak{g}}_s$; that is, $\mathfrak{d}_s = \sum_{k=1}^{K} \lambda_{k,s}^* \tilde{\mathfrak{g}}_{k,s}$. |

**Theorem B.1.** Assume that $\boldsymbol{f} = \{f_k\}_{k \in [K]}$ are l-Lipschitz continuous and L-Lipschitz smooth, and that the step-size $\eta^t$ satisfies the following three conditions: (i) $\eta^t \in (0, \frac{1}{2L}]$, (ii) $\lim_{T \to \infty} \sum_{t=0}^{T} \eta^t \to \infty$ and (iii) $\lim_{T \to \infty} \sum_{t=0}^{T} \eta^t \sigma_t < \infty$; where $\sigma_t^2 = \mathbf{E}[\|\tilde{\mathbf{g}}\boldsymbol{\lambda}^* - \tilde{\mathbf{g}}_s \boldsymbol{\lambda}_s^*\|]^2$ is the variance of stochastic common descent direction. Then

$$\lim_{T \to \infty} \min_{t=0,\ldots,T} \mathbf{E}[\|\mathfrak{d}^t\|] \to 0. \tag{36}$$

*Proof.* Since orthogonal vectors $\{\tilde{\mathfrak{g}}_k\}_{k \in [K]}$ span the same $K$-dimensional space as that spanned by gradient vectors $\{\mathfrak{g}_k\}_{k \in [K]}$, then

$$\exists \{\lambda_k'\}_{k \in [K]} \quad \text{s.t.} \quad \mathfrak{d} = \sum_{k=1}^{K} \lambda_k^* \tilde{\mathfrak{g}}_k = \sum_{k=1}^{K} \lambda_k' \mathfrak{g}_k = \mathbf{g}\boldsymbol{\lambda}'. \tag{37}$$

Similarly, for the stochastic gradients we have

$$\exists \{\lambda_{k,s}'\}_{k \in [K]} \quad \text{s.t.} \quad \mathfrak{d}_s = \sum_{k=1}^{K} \lambda_{k,s}^* \tilde{\mathfrak{g}}_{k,s} = \sum_{k=1}^{K} \lambda_{k,s}' \mathfrak{g}_{k,s} = \mathbf{g}_s \boldsymbol{\lambda}_s'. \tag{38}$$

Define $\Delta_t = \mathbf{g}\boldsymbol{\lambda}' - \mathbf{g}_s\boldsymbol{\lambda}_s' = \tilde{\mathbf{g}}\boldsymbol{\lambda}^* - \tilde{\mathbf{g}}_s\boldsymbol{\lambda}_s^*$, where the last equality is due to the definitions in Equations (37) and (38).

We can find an upper bound for $\boldsymbol{f}(\boldsymbol{\theta}^{t+1})$ as follows

$$\boldsymbol{f}(\boldsymbol{\theta}^{t+1}) = \boldsymbol{f}(\boldsymbol{\theta}^t - \eta^t \mathfrak{d}_t) \tag{39}$$

$$= \boldsymbol{f}(\boldsymbol{\theta}^t - \eta^t \sum_{k=1}^{K} \lambda_{k,s}^* \tilde{\mathfrak{g}}_{k,s}) \tag{40}$$

$$= \boldsymbol{f}(\boldsymbol{\theta}^t - \eta^t \mathbf{g}_s \boldsymbol{\lambda}_s') \tag{41}$$

$$\leq \boldsymbol{f}(\boldsymbol{\theta}^t) - \eta^t \mathbf{g}^T \mathbf{g}_s^T \boldsymbol{\lambda}_s' + \frac{L(\eta^t)^2}{2} \|\mathbf{g}_s^T \boldsymbol{\lambda}_s'\|^2 \tag{42}$$

$$\leq \boldsymbol{f}(\boldsymbol{\theta}^t) - \eta^t \mathbf{g}^T \mathbf{g}^T \boldsymbol{\lambda}' + L(\eta^t)^2 \|\mathbf{g}^T \boldsymbol{\lambda}'\|^2 + \eta^t \mathbf{g}^T \Delta_t + L(\eta^t)^2 \|\Delta_t\|^2 \tag{43}$$

$$\leq \boldsymbol{f}(\boldsymbol{\theta}^t) - \eta^t (1 - L\eta^t) \|\mathbf{g}^T \boldsymbol{\lambda}'\|^2 + l\eta^t \|\Delta_t\| + L(\eta^t)^2 \|\Delta_t\|^2, \tag{44}$$

where (40) uses stochastic gradients in the updating rule of DQN-Fed, (41) is obtained from the definition in (38), (42) holds following the quadratic bound for smooth functions $\boldsymbol{f} = \{f_k\}_{k \in [K]}$, and lastly (44) holds considering the Lipschits continuity of $\boldsymbol{f} = \{f_k\}_{k \in [K]}$.

Assuming $\eta^t \in (0, \frac{1}{2L}]$ and taking expectation from both sides, we obtain:

$$\min_{t=0,\ldots,T} \mathbf{E}[\|\mathfrak{d}_t\|] \leq \frac{\boldsymbol{f}(\boldsymbol{\theta}_0) - \mathbf{E}[\boldsymbol{f}(\boldsymbol{\theta}^{t+1})] + \sum_{t=0}^{T} \eta^t (l\sigma_t + L\eta^t \sigma_t^2)}{\frac{1}{2} \sum_{t=0}^{T} \eta^t}. \tag{45}$$

Using the assumptions (i) $\lim_{T \to \infty} \sum_{j=0}^{T} \eta^t \to \infty$, and (ii) $\lim_{T \to \infty} \sum_{t=0}^{T} \eta^t \sigma_t < \infty$, the theorem will be concluded. Note that *vanishing* $\mathfrak{d}^t$ implies reaching to a Pareto-stationary point of original MoM problem. Yet, the convergence rate is different in different scenarios as we see in the following theorems. $\square$

### B.1.1 Discussing the assumptions

● **The assumptions over the local loss functions:** The two assumptions l-Lipschitz continuous and L-Lipschitz smooth over the local loss functions are two standard assumptions in FL papers providing some sorts of convergence guarantee (Li et al., 2019b).

● **The assumptions over the step-size:** The three assumptions we enforced over the step-size could be easily satisfied as explained in the sequel. For instance, one can pick $\eta^t = \kappa_1 \frac{1}{t}$ for some constant $\kappa_1$ such that

$\eta^t \in (0, \frac{1}{2L}]$ is satisfied. Then even if $\sigma_t$ has a extremely loose upper-bound, let's say $\sigma_t < \frac{\kappa_2}{t^\epsilon}$ for a small $\epsilon \in \mathbb{R}_+$ and a constant number $\kappa_2$, then all the three assumptions over the step-size in the theorem will be satisfied. Note that the convergence rate of DQN-Fed depends on how fast $\sigma_t$ diminishes which depends on how heterogeneous the users are.

## B.2 Case 2: $E > 1$ & local GD

The notations used in this subsection are elaborated in Table 8.

Table 8: Notations used in the Theorem B.2 for $E > 1$ and local GD.

| Notation | Description |
|---|---|
| $\boldsymbol{\theta}_{(k,E)^t}$ | Updated weight for client $k$ after $E$ local epochs at the $t$-th round of FL. |
| $\mathfrak{g}_{k,E}$ | $\mathfrak{g}_{k,E} = \boldsymbol{\theta}^t - \boldsymbol{\theta}_{(k,E)^t}$; that is, the update vector of client $k$ after $E$ local epochs. |
| $\mathbf{\mathfrak{g}}_E$ | Matrix of update vectors $[\mathfrak{g}_{1,E}, \ldots, \mathfrak{g}_{K,e}]$. |
| $\tilde{\mathfrak{g}}_{k,E}$ | Update vector of client $k$ after orthogonalization process. |
| $\tilde{\mathbf{\mathfrak{g}}}_E$ | Matrix of orthogonalized update vectors $[\tilde{\mathfrak{g}}_{1,E}, \ldots, \tilde{\mathfrak{g}}_{K,e}]$. |
| $\lambda^*_{k,E}$ | Optimum weights obtained from Equation (12) using $\tilde{\mathbf{\mathfrak{g}}}_E$. |
| $\mathfrak{d}_e$ | Optimum direction obtained using $\tilde{\mathbf{\mathfrak{g}}}_E$; that is, $\mathfrak{d}_E = \sum_{k=1}^{K} \lambda^*_{k,E} \tilde{\mathfrak{g}}_{k,E}$. |

**Theorem B.2.** Assume that $\boldsymbol{f} = \{f_k\}_{k \in [K]}$ are l-Lipschitz continuous and L-Lipschitz smooth. Denote by $\eta^t$ and $\eta$ the global and local learning rate, respectively. Also, define $\zeta^t = \|\boldsymbol{\lambda}^* - \boldsymbol{\lambda}^*_E\|$, where $\boldsymbol{\lambda}^*_E$ is the optimum weights obtained from pseudo-gradients after $E$ local epochs. Then,

$$\lim_{T \to \infty} \min_{t=0,\ldots,T} \|\mathfrak{d}^t\| \to 0, \tag{46}$$

if the following conditions are satisfied: (i) $\eta^t \in (0, \frac{1}{2L}]$, (ii) $\lim_{T \to \infty} \sum_{t=0}^{T} \eta^t \to \infty$ and (iii) $\lim_{t \to \infty} \eta^t \to 0$, (iv) $\lim_{t \to \infty} \eta \to 0$, and (v) $\lim_{t \to \infty} \zeta^t \to 0$.

*Proof.* As discussed in the proof of Theorem B.1, we can write

$$\exists \{\lambda'_k\}_{k \in [K]} \quad \text{s.t.} \quad \mathfrak{d} = \sum_{k=1}^{K} \lambda^*_k \tilde{\mathfrak{g}}_k = \sum_{k=1}^{K} \lambda'_k \mathfrak{g}_k = \mathfrak{g} \boldsymbol{\lambda}', \tag{47}$$

$$\exists \{\lambda'_{k,E}\}_{k \in [K]} \quad \text{s.t.} \quad \mathfrak{d}_E = \sum_{k=1}^{K} \lambda^*_{k,E} \tilde{\mathfrak{g}}_{k,E} = \sum_{k=1}^{K} \lambda'_{k,E} \mathfrak{g}_{k,E} = \mathbf{\mathfrak{g}}_E \boldsymbol{\lambda}'_E. \tag{48}$$

To prove Theorem B.2, we first introduce a lemma whose proof is provided in Appendix C.

**Lemma B.3.** Using the notations used in Theorem B.2, and assumming that $\boldsymbol{f} = \{f_k\}_{k \in [K]}$ are L-Lipschitz smooth, we have $\|\mathfrak{g}_{k,E} - \mathfrak{g}_k\| \leq \eta e l$.

Using Lemma B.3, we have

$$\|\mathfrak{d} - \mathfrak{d}_{\boldsymbol{E}}\| = \|\tilde{\mathfrak{g}} \boldsymbol{\lambda}^* - \tilde{\mathbf{\mathfrak{g}}}_E \boldsymbol{\lambda}^*_E\| \leq \|\tilde{\mathfrak{g}} \boldsymbol{\lambda}^* - \tilde{\mathfrak{g}} \boldsymbol{\lambda}^*_E\| + \|\tilde{\mathfrak{g}} \boldsymbol{\lambda}^*_E - \tilde{\mathbf{\mathfrak{g}}}_E \boldsymbol{\lambda}^*_E\| \tag{49}$$

$$\leq \|\tilde{\mathfrak{g}}\| \|\boldsymbol{\lambda}^* - \boldsymbol{\lambda}^*_E\| + \|\mathfrak{g} \boldsymbol{\lambda}'_E - \mathbf{\mathfrak{g}}_E \boldsymbol{\lambda}'_E\| \tag{50}$$

$$\leq \|\tilde{\mathfrak{g}}\| \|\boldsymbol{\lambda}^* - \boldsymbol{\lambda}^*_E\| + \eta e l \tag{51}$$

$$\leq \zeta^t l \sqrt{K} + \eta e l, \tag{52}$$

where Equation (49) follows triangular inequality, Equation (50) is obtained from Equations (47) and (48), and Equation (51) uses Lemma B.3.

As seen, if $\lim_{t \to \infty} \eta \to 0$, and $\lim_{t \to \infty} \zeta^t \to 0$, then $\|\mathfrak{d} - \mathfrak{d}_E\| \to 0$. Now, by writing the quadratic upper bound we obtain:

$$
\boldsymbol{f}(\boldsymbol{\theta}^{t+1}) \leq \boldsymbol{f}(\boldsymbol{\theta}^t) - \eta^t \mathbf{g}^T \mathbf{g}_E^T \boldsymbol{\lambda}'_e + \frac{L(\eta^t)^2}{2} \|\mathbf{g}_E^T \boldsymbol{\lambda}'_e\|^2 \tag{53}
$$

$$
\leq \boldsymbol{f}(\boldsymbol{\theta}^t) - \eta^t \mathbf{g}^T \mathbf{g}^T \boldsymbol{\lambda}' + L(\eta^t)^2 \|\mathbf{g}^T \boldsymbol{\lambda}'\|^2 + \eta^t \mathbf{g}^T (\mathfrak{d} - \mathfrak{d}_E) + L(\eta^t)^2 \|\mathfrak{d} - \mathfrak{d}_E\|^2 \tag{54}
$$

$$
\leq \boldsymbol{f}(\boldsymbol{\theta}^t) - \eta^t(1 - L\eta^t)\|\mathbf{g}^T \boldsymbol{\lambda}'\|^2 + l\eta^t \|\mathfrak{d} - \mathfrak{d}_E\| + L(\eta^t)^2 \|\mathfrak{d} - \mathfrak{d}_E\|^2. \tag{55}
$$

Noting that $\eta^t \in (0, \frac{1}{2L}]$, and utilizing telescoping yields

$$
\min_{t=0,\ldots,T} \|\mathfrak{d}_t\| \leq \frac{\boldsymbol{f}(\boldsymbol{\theta}_0) - \boldsymbol{f}(\boldsymbol{\theta}^{t+1}) + \sum_{t=0}^{T} \eta^t (l\|\mathfrak{d} - \mathfrak{d}_E\| + L\eta^t \|\mathfrak{d} - \mathfrak{d}_E\|^2)}{\frac{1}{2} \sum_{t=0}^{T} \eta^t}. \tag{56}
$$

Using $\|\mathfrak{d} - \mathfrak{d}_E\| \to 0$, the Theorem B.2 is concluded. $\qquad \square$

### B.3   Case 3: $E = 1$ & local GD

Denote by $\vartheta$ the Pareto-stationary solution set of minimization problem $\arg \min_{\boldsymbol{\theta}} \boldsymbol{f}(\boldsymbol{\theta})$. Then, define $\boldsymbol{\theta}^* = \arg \min_{\boldsymbol{\theta} \in \vartheta} \|\boldsymbol{\theta}^t - \boldsymbol{\theta}\|^2$.

**Theorem B.4.** Assume that $\boldsymbol{f} = \{f_k\}_{k \in [K]}$ are l-Lipschitz continuous and $\sigma$-convex, and that the step-size $\eta^t$ satisfies the following two conditions: (i) $\lim_{t \to \infty} \sum_{j=0}^{t} \eta_j \to \infty$ and (ii) $\lim_{t \to \infty} \sum_{j=0}^{t} \eta_j^2 < \infty$. Then almost surely $\boldsymbol{\theta}^t \to \boldsymbol{\theta}^*$; that is,

$$
\mathbb{P}\left(\lim_{t \to \infty} (\boldsymbol{\theta}^t - \boldsymbol{\theta}^*) = 0\right) = 1, \tag{57}
$$

where $\mathbb{P}(E)$ denotes the probability of event $E$.

*Proof.* The proof is inspired from Mercier et al. (2018). Without loss of generality, we assume that all users participate in all rounds.

Based on the definition of $\boldsymbol{\theta}^*$ we can say

$$
\|\boldsymbol{\theta}^{t+1} - \boldsymbol{\theta}^*_{t+1}\|^2 \leq \|\boldsymbol{\theta}^{t+1} - \boldsymbol{\theta}^*_t\|^2 = \|\boldsymbol{\theta}^t - \eta^t \mathfrak{d}_t - \boldsymbol{\theta}^*_t\|^2 \tag{58}
$$

$$
= \|\boldsymbol{\theta}^t - \boldsymbol{\theta}^*_t\|^2 - 2\eta^t(\boldsymbol{\theta}^t - \boldsymbol{\theta}^*_t) \cdot \mathfrak{d}_t + (\eta^t)^2 \|\mathfrak{d}_t\|^2. \tag{59}
$$

To bound the third term in Equation (59), we note that from Equation (34), we have:

$$
(\eta^t)^2 \|\mathfrak{d}_t\|^2 = \frac{(\eta^t)^2}{\sum_{k=1}^{K} \frac{1}{\|\tilde{\mathfrak{g}}_k\|^2}} \leq \frac{(\eta^t)^2 l^2}{K}. \tag{60}
$$

To bound the second term, first note that since orthogonal vectors $\{\tilde{\mathfrak{g}}_k\}_{k \in [K]}$ span the same $K$-dimensional space as that spanned by gradient vectors $\{\mathfrak{g}_k\}_{k \in [K]}$, then

$$
\exists \{\lambda'_k\}_{k \in [K]} \text{ s.t. } \mathfrak{d} = \sum_{k=1}^{K} \lambda^*_k \tilde{\mathfrak{g}}_k = \sum_{k=1}^{K} \lambda'_k \mathfrak{g}_k. \tag{61}
$$

Using Equation (61) and the $\sigma$-convexity of $\{f_k\}_{k\in[K]}$ we obtain

$$(\boldsymbol{\theta}^t - \boldsymbol{\theta}_t^*) \cdot \mathfrak{d}_t = (\boldsymbol{\theta}^t - \boldsymbol{\theta}_t^*) \cdot \sum_{k=1}^{K} \lambda_k^* \tilde{\mathfrak{g}}_k \tag{62}$$

$$= (\boldsymbol{\theta}^t - \boldsymbol{\theta}_t^*) \cdot \sum_{k=1}^{K} \lambda_k' \mathfrak{g}_k \tag{63}$$

$$\geq \sum_{k=1}^{K} \lambda_k' \left( f_k(\boldsymbol{\theta}^t) - f_k(\boldsymbol{\theta}_t^*) \right) + \sigma \frac{\|\boldsymbol{\theta}^t - \boldsymbol{\theta}_t^*\|^2}{2} \tag{64}$$

$$\geq \frac{\lambda_\alpha' M}{2} \|\boldsymbol{\theta}^t - \boldsymbol{\theta}_t^*\|^2 + \sigma \frac{\|\boldsymbol{\theta}^t - \boldsymbol{\theta}_t^*\|^2}{2} \tag{65}$$

$$= \frac{\lambda_\alpha' M + \sigma}{2} \|\boldsymbol{\theta}^t - \boldsymbol{\theta}_t^*\|^2. \tag{66}$$

Now, we return back to Equation (59) and find the conditional expectation w.r.t. $\boldsymbol{\theta}^t$ as follows

$$\mathbf{E}[\|\boldsymbol{\theta}^{t+1} - \boldsymbol{\theta}_{t+1}^*\|^2 \mid \boldsymbol{\theta}^t] \leq (1 - \eta^t \mathbf{E}[\lambda_\alpha' M + \sigma|\boldsymbol{\theta}^t])\|\boldsymbol{\theta}^t - \boldsymbol{\theta}_t^*\|^2 + \frac{(\eta^t)^2 l^2}{K}. \tag{67}$$

Assume that $\mathbf{E}[\lambda_\alpha' M + \sigma|\boldsymbol{\theta}^t] \geq c$, taking another expectation we obtain:

$$\mathbf{E}[\|\boldsymbol{\theta}^{t+1} - \boldsymbol{\theta}_{t+1}^*\|^2] \leq (1 - \eta^t c)\mathbf{E}[\|\boldsymbol{\theta}^t - \boldsymbol{\theta}_t^*\|^2] + \frac{(\eta^t)^2 l^2}{K}, \tag{68}$$

which is a recursive expression. By solving Equation (68) we obtain

$$\mathbf{E}[\|\boldsymbol{\theta}^{t+1} - \boldsymbol{\theta}_{t+1}^*\|^2] \leq \underbrace{\prod_{j=0}^{t}(1 - \eta_j c)\mathbf{E}[\|\boldsymbol{\theta}_0 - \boldsymbol{\theta}_0^*\|^2]}_{\text{First term}} + \underbrace{\sum_{m=1}^{t} \frac{\prod_{j=1}^{t}(1 - \eta_j c)\eta_m^2 l^2}{K \prod_{j=1}^{m}(1 - \eta_j c)}}_{\text{Second term}}. \tag{69}$$

It is observed that if the limit of both First term and Second term in Equation (69) go to zero, then $\mathbf{E}[\|\boldsymbol{\theta}^{t+1} - \boldsymbol{\theta}_{t+1}^*\|^2] \to 0$. For the First term, from the arithmetic-geometric mean inequality we have

$$\lim_{t\to\infty} \prod_{j=0}^{t}(1 - \eta_j c) \leq \lim_{t\to\infty} \left( \frac{\sum_{j=0}^{t}(1 - \eta_j c)}{t} \right)^t = \lim_{t\to\infty} \left( 1 - c\frac{\sum_{j=0}^{t} \eta_j}{t} \right)^t \tag{70}$$

$$= \lim_{t\to\infty} e^{-c\sum_{j=0}^{t} \eta_j}. \tag{71}$$

From Equation (71) it is seen that if $\lim_{t\to\infty} \sum_{j=0}^{t} \eta_j \to \infty$, then the First term is also converges to zero as $t \to \infty$.

On the other hand, consider the Second term in Equation (69). Obviously, if $\lim_{t\to\infty} \sum_{j=0}^{t} \eta_j^2 < \infty$, then the Second term converges to zero as $t \to \infty$.

Hence, if (i) $\lim_{t\to\infty} \sum_{j=0}^{t} \eta_j \to \infty$ and (ii) $\lim_{t\to\infty} \sum_{j=0}^{t} \eta_j^2 < \infty$, then $\mathbf{E}[\|\boldsymbol{\theta}^{t+1} - \boldsymbol{\theta}_{t+1}^*\|^2] \to 0$. Consequently, based on standard supermartingale (Mercier et al., 2018), we have

$$\mathbb{P}\left( \lim_{t\to\infty} \left( \boldsymbol{\theta}^t - \boldsymbol{\theta}^* \right) = 0 \right) = 1. \tag{72}$$

$\square$

## C  Proof of Lemma B.3

*Proof.*

$$\mathfrak{g}_{k,E} = \boldsymbol{\theta}^t - \boldsymbol{\theta}_{(k,E)^t} = (\boldsymbol{\theta}^t - \boldsymbol{\theta}_{(k,1)^t}) + (\boldsymbol{\theta}_{(k,1)^t} - \boldsymbol{\theta}_{(k,2)^t}) + \cdots + (\boldsymbol{\theta}_{(k,e-1)^t} - \boldsymbol{\theta}_{(k,E)^t}) \tag{73}$$

$$= \mathfrak{g}_k(\boldsymbol{\theta}^t) + \eta\mathfrak{g}_{k,1} + \cdots + \eta\mathfrak{g}_{k,e-1}. \tag{74}$$

Hence,

$$\|\mathfrak{g}_{k,E} - \mathfrak{g}_k\| = \|\eta \sum_{j=1}^{e} \mathfrak{g}_{k,j}\| \le \eta \sum_{j=1}^{e} \|\mathfrak{g}_{k,j}\| \le \eta e l. \tag{75}$$

□

## D    Proof of Theorem Theorem 4.6

Throughout the proofs in this section, we frequently use the triangular inequality for two vectors $\mathbf{v}$ and $\mathbf{u}$: $\|\mathbf{v} \pm \mathbf{u}\| \le \|\mathbf{v}\| + \|\mathbf{u}\|$.

Our goal is to prove the theorem by deriving a recursive relation for the distance between the optimal global model $\boldsymbol{\theta}^{\text{opt}}$ and the global model at the $t$-th round $\boldsymbol{\theta}_t$, specifically $\|\boldsymbol{\theta}_t - \boldsymbol{\theta}^{\text{opt}}\|$. First, noting that $\boldsymbol{\theta}_{t+1} = \boldsymbol{\theta}_t + \eta_t \mathbf{B}_t^{-1} \tilde{\mathbf{g}}_t$, we have

$$\begin{aligned}
\|\boldsymbol{\theta}_{t+1} - \boldsymbol{\theta}^{\text{opt}}\| &= \|\boldsymbol{\theta}_t - \eta_t \mathbf{B}_t^{-1} \tilde{\mathbf{g}}_t - \boldsymbol{\theta}^{\text{opt}}\| \\
&\le \underbrace{\|\boldsymbol{\theta}_t - \eta_t \mathbf{H}_t^{-1} \mathbf{g}_t - \boldsymbol{\theta}^{\text{opt}}\|}_{M_1} \\
&\quad + \eta_t \underbrace{\|\mathbf{H}_t^{-1} \mathbf{g}_t - \mathbf{B}_t^{-1} \tilde{\mathbf{g}}_t\|}_{M_2}.
\end{aligned} \tag{76}$$

To bound $M_1$, we use the results in Polyak & Tremba (2020). In particular, define $t_0 = \max\left\{0, \left\lceil \frac{2L}{\lambda^2 \|\mathfrak{d}^0\|} \right\rceil - 2\right\}$, $\gamma = \frac{L}{2\lambda^2} \|\mathfrak{d}^0\| - \frac{t_0}{4}$; then,

$$M_1 \le \begin{cases} \frac{\lambda}{L}(t_0 - t + \frac{2\gamma}{1-\gamma}), & t \le t_0 \\ \frac{2\lambda\gamma^{2^{t-t_0}}}{L(1-\gamma^{2^{t-t_0}})}, & t > t_0 \end{cases} \tag{77}$$

Next, we bound $M_2$ in the sequel.

$$M_2 \le \|\mathbf{H}_t^{-1} \mathbf{g}_t - \mathbf{B}_t^{-1} \mathbf{g}_t\| + \|\mathbf{B}_t^{-1} \mathbf{g}_t - \mathbf{B}_t^{-1} \tilde{\mathbf{g}}_t\| \tag{78}$$
$$\le \|\mathbf{H}_t^{-1} - \mathbf{B}_t^{-1}\| \|\mathbf{g}_t\| + \|\mathbf{B}_t^{-1}\| \|\mathbf{g}_t - \tilde{\mathbf{g}}_t\|, \tag{79}$$

where in (78) we used triangular inequality. Note that using the assumption 2, we have $\|\mathbf{H}_t^{-1} - \mathbf{B}_t^{-1}\| \le \delta \|\mathbf{H}_t^{-1}\|$, and by $\lambda$-strong convexity of the loss function, we have $\|\mathbf{H}_t^{-1}\| \le \frac{1}{\lambda}$. Hence,

$$\|\mathbf{H}_t^{-1} - \mathbf{B}_t^{-1}\| \le \frac{\delta}{\lambda}. \tag{80}$$

In addition, the $L$-smoothness of the global loss function yields

$$\|\mathbf{g}_t\| \le L \|\boldsymbol{\theta}_t - \boldsymbol{\theta}^{\text{opt}}\|. \tag{81}$$

Hence, from (80) and (81), the first term in (79) could be bounded. Now, to bound the second term in (79), note that

$$\|\mathbf{B}_t^{-1}\| \le \|\mathbf{B}_t^{-1} - \mathbf{H}_t^{-1}\| + \|\mathbf{H}_t^{-1}\| \tag{82}$$
$$\le \frac{\delta}{\lambda} + \frac{1}{\lambda} = \frac{\delta+1}{\lambda}. \tag{83}$$

Using (80), (81) and (82) in the inequality (78) we obtain

$$M_2 \le \frac{L\delta}{\lambda} \|\boldsymbol{\theta}_t - \boldsymbol{\theta}^{\text{opt}}\| + \frac{\delta+1}{\lambda} \|\mathbf{g}_t - \tilde{\mathbf{g}}_t\|. \tag{84}$$

Next, we have (note that $\eta_t \leq 1$)

$$\|\boldsymbol{\theta}_{t+1} - \boldsymbol{\theta}^{\text{opt}}\| \leq \begin{cases} \frac{L\delta}{\lambda}\|\boldsymbol{\theta}_t - \boldsymbol{\theta}^{\text{opt}}\| + A_0, & t \leq t_0 \\ \frac{L\delta}{\lambda}\|\boldsymbol{\theta}_t - \boldsymbol{\theta}^{\text{opt}}\| + A_1, & t > t_0 \end{cases} \tag{85a}$$

$$\text{where} \quad A_0 = \frac{\lambda}{L}(t_0 - t + \frac{2\gamma}{1-\gamma}), \tag{85b}$$

$$\text{and} \quad A_1 = \frac{2\lambda\gamma^{2^{t-t_0}}}{L(1 - \gamma^{2^{t-t_0}})}. \tag{85c}$$

Applying (85) recursively yields

$$\|\boldsymbol{\theta}_t - \boldsymbol{\theta}^{\text{opt}}\| \leq \begin{cases} \left(\frac{L\delta}{\lambda}\right)^t \|\boldsymbol{\theta}_0 - \boldsymbol{\theta}^{\text{opt}}\| + A_0', & t \leq t_0 \\ \left(\frac{L\delta}{\lambda}\right)^t \|\boldsymbol{\theta}_0 - \boldsymbol{\theta}^{\text{opt}}\| + A_1', & t > t_0 \end{cases} \tag{86a}$$

$$\text{where} \quad A_0' = \frac{\left(\frac{L\delta}{\lambda}\right)^t - 1}{\frac{L\delta}{\lambda} - 1} A_0, \tag{86b}$$

$$\text{and} \quad A_1' = \frac{\left(\frac{L\delta}{\lambda}\right)^t - 1}{\frac{L\delta}{\lambda} - 1} A_1. \tag{86c}$$

## E    Proof of Corollary 4.7

As per Theorem (4.6), if $\left\|\boldsymbol{\theta}_0 - \boldsymbol{\theta}^{\text{opt}}\right\| < \frac{A_1'}{\left(\frac{L\delta}{\lambda}\right)^t}$, then

$$\|\boldsymbol{\theta}_t - \boldsymbol{\theta}^{\text{opt}}\| \leq \left(\frac{L\delta}{\lambda}\right)^t \frac{A_1'}{\left(\frac{L\delta}{\lambda}\right)^t} + A_1' = 2A_1'. \tag{87}$$

Hence, to find $T_\epsilon$, we shall have

$$2A_1' \leq \epsilon \tag{88}$$

$$\Leftrightarrow 2\frac{\left(\frac{L\delta}{\lambda}\right)^t - 1}{\frac{L\delta}{\lambda} - 1} \underbrace{\left[\frac{2\lambda\gamma^{2^{t-t_0}}}{L(1 - \gamma^{2^{t-t_0}})}\right]}_{\text{diminishing term}} \leq \epsilon. \tag{89}$$

Since $\left(\frac{L\delta}{\lambda}\right) < 1$, as $t$ becomes larger, $\left(\frac{L\delta}{\lambda}\right)^t \approx 0$, and therefore $2\frac{\left(\frac{L\delta}{\lambda}\right)^t - 1}{\frac{L\delta}{\lambda} - 1} \approx \frac{2}{1 - \frac{L\delta}{\lambda}}$. In addition, since $\gamma \in [0, \frac{1}{2}]$, for the large values of $t$, $\frac{2\lambda\gamma^{2^{t-t_0}}}{L(1-\gamma^{2^{t-t_0}})} \approx \frac{2\lambda\gamma^{2^{t-t_0}}}{L}$. Thus, by inverting the inequality (89), and then taking log from both sides we have

$$\log(\frac{1}{\epsilon}) \leq -\log(\frac{4\lambda}{L - \frac{L^2\delta}{\lambda}}) - 2^{t-t_0}\log(\gamma). \tag{90}$$

Note that $\log(\gamma) < 0$, and therefore the second term on the RHS of (90) is positive. Also, since $-\log(\frac{4\lambda}{L-\frac{L^2\delta}{\lambda}}) \ll -2^{t-t_0}\log(\gamma)$, then

$$T_\epsilon \leq \mathcal{O}\left(\log\log\frac{1}{\epsilon}\right). \tag{91}$$

# F   Proof of Corollary 4.8

Based on Theorem (4.6), if $\left\|\boldsymbol{\theta}_0 - \boldsymbol{\theta}^{\text{opt}}\right\| \geq \frac{A_1'}{\left(\frac{L\delta}{\lambda}\right)^t}$, then

$$\left\|\boldsymbol{\theta}_t - \boldsymbol{\theta}^{\text{opt}}\right\| \leq \left(\frac{L\delta}{\lambda}\right)^t \left\|\boldsymbol{\theta}_0 - \boldsymbol{\theta}^{\text{opt}}\right\| + \left(\frac{L\delta}{\lambda}\right)^t \left\|\boldsymbol{\theta}_0 - \boldsymbol{\theta}^{\text{opt}}\right\| \tag{92}$$

$$\leq 2\left(\frac{L\delta}{\lambda}\right)^t \left\|\boldsymbol{\theta}_0 - \boldsymbol{\theta}^{\text{opt}}\right\|. \tag{93}$$

Thus, to find $T_\epsilon$, we shall have

$$2\left(\frac{L\delta}{\lambda}\right)^{T_\epsilon} \left\|\boldsymbol{\theta}_0 - \boldsymbol{\theta}^{\text{opt}}\right\| \leq \epsilon \tag{94}$$

$$\Leftrightarrow \quad T_\epsilon \log(\frac{\lambda}{L\delta}) \geq \log\left(\frac{2\left\|\boldsymbol{\theta}_0 - \boldsymbol{\theta}^{\text{opt}}\right\|}{\epsilon}\right). \tag{95}$$

Hence

$$T_\epsilon = \mathcal{O}\left(\frac{1}{\log(\frac{\lambda}{L\delta})} \log\frac{1}{\epsilon}\right). \tag{96}$$

# G   Additional datasets

In this section, we assess the performance of DQN-Fed against several benchmarks using additional datasets, namely Fashion MNIST, CINIC-10, and TinyImageNet. The corresponding results for each dataset are detailed in Appendices G.1 to G.3.

## G.1   Fashion MNIST

Fashion MNIST (Xiao et al., 2017) is an extension of MNIST dataset (LeCun et al., 1998) with images resized to $32 \times 32$ pixels.

We use a fully-connected neural network with 2 hidden layers, and use the same setting as that used in Li et al. (2019a) for our experiments. We set $E = 1$ and use full batchsize, and use $\eta = 0.1$. Then, we conduct 300 rounds of communications. For the benchmarks, we use the same as those we used for CIFAR-10 experiments. The results are reported in Table 9.

By observing the three different classes reported in Table 9, we observe that the fairness level attained in DQN-Fed is not limited to a dominate class.

Table 9: Test accuracy on Fashion MNIST. The reported results are averaged over 5 different seeds.

| ALGORITHM | $\bar{a}$ | $\sigma_a$ | SHIRT | PULLOVER | T-SHIRT |
|---|---|---|---|---|---|
| FEDAVG | **80.42** | 3.39 | 64.26 | 87.00 | 89.90 |
| Q-FFL | 78.53 | **2.27** | 71.29 | 81.46 | 82.86 |
| FEDMGDA+ | 79.29 | 2.53 | 72.46 | 79.74 | 85.66 |
| FEDHEAL | 80.22 | 3.41 | 63.71 | 86.87 | 89.94 |
| DQN-FED | 81.27 | 2.31 | **72.57** | **88.21** | **90.99** |

## G.2   CINIC-10

CINIC-10 (Darlow et al., 2018) has 4.5 times as many images as those in CIFAR-10 dataset (270,000 sample images in total). In fact, it is obtained from ImageNet and CIFAR-10 datasets. As a result, this dataset fits FL

scenarios since the constituent elements of CINIC-10 are not drawn from the same distribution. Furthermore, we add more non-iidness to the dataset by distributing the data among the clients using Dirichlet allocation with $\beta = 0.5$.

For the model, we use ResNet-18 with group normalization, and set $\eta = 0.01$. There are 200 communication rounds in which all the clients participate with $E = 1$. Also, $K = 50$. Results are reported in Table 10.

Table 10: Test accuracy on CINIC-10. The reported results are averaged over 5 different seeds.

| ALGORITHM | $\bar{a}$ | $\sigma_a$ | WORST 10% | BEST 10% |
|---|---|---|---|---|
| Q-FFL | 86.57 | 14.91 | 57.70 | 100.00 |
| DITTO | 86.31 | 15.14 | 56.91 | 100.00 |
| FEDLF | 86.49 | 15.12 | 57.62 | 100.00 |
| TERM | 86.40 | 15.10 | 57.30 | 100.00 |
| DQN-FED | **87.34** | **14.85** | **57.88** | 99.99 |

### G.3 TinyImageNet

Tiny-ImageNet (Le & Yang, 2015) is a subset of ImageNet with 100k samples of 200 classes. We distribute the dataset among $K = 20$ clients using Dirichlet allocation with $\beta = 0.05$

We use ResNet-18 with group normalization, and set $\eta = 0.02$. There are 400 communication rounds in which all the clients participate with $E = 1$. The results are reported in Table 11.

Table 11: Test accuracy on TinyImageNet. The reported results are averaged over 5 different seeds.

| ALGORITHM | $\bar{a}$ | $\sigma_a$ | WORST 10% | BEST 10% |
|---|---|---|---|---|
| Q-FFL | 18.90 | 3.20 | 13.12 | 23.72 |
| FEDLF | 16.55 | **2.38** | 12.40 | 20.25 |
| TERM | 16.41 | 2.77 | 11.52 | 21.02 |
| FEDMGDA+ | 14.00 | 2.71 | 9.88 | 19.21 |
| DQN-FED | **19.05** | 2.35 | **13.24** | **23.58** |

## H Experiments details, tuning hyper-parameters

For all benchmark methods, we conducted a grid-search to identify the optimal hyper-parameters for the underlying algorithms. The parameters tested for each method are outlined below:

- **q-FFL:** $q \in \{0, 0.001, 0.01, 0.1, 1, 2, 5, 10\}$.
- **TERM:** $t \in \{0.1, 0.5, 1, 2, 5\}$.
- **FedLF:** $\eta^t \in \{0.01, 0.05, 0.1, 0.5, 1\}$.
- **Ditto:** $\lambda \in \{0.01, 0.05, 0.1, 0.5, 1, 2, 5\}$.
- **FedMGDA+:** $\epsilon \in \{0.01, 0.05, 0.1, 0.5, 1\}$.
- **FedHEAL:** $(\alpha, \beta) = \{(0.5, 0.5)\}$, $(\gamma_s, \gamma_c) = \{(0.5, 0.9)\}$.

## I Integration with a Label Noise Correction method

### I.1 Understanding Label Noise in FL

Label noise in FL refers to inaccuracies or errors in the ground truth labels of training data, which can arise due to various factors. These errors may occur during data collection, annotation, or transmission, leading

to incorrect or noisy labels. In wireless FL settings, label noise can also result from transmission errors over unreliable communication channels, where data packets may be corrupted or lost, causing mislabeling (Hamidi et al., 2022). Addressing label noise is critical, as it can significantly degrade the performance and reliability of FL models, necessitating the development of robust techniques to mitigate its impact.

Addressing label noise in FL presents unique challenges due to its reliance on decentralized data sources, where participants may have limited control over label quality in remote environments. Mitigating label noise in FL requires the development of robust models and FL algorithms capable of adapting to inaccuracies in the labels. This adaptation is essential for maintaining model performance and reliability in real-world FL scenarios where label noise is prevalent.

## I.2 Robustness of Fair FL Algorithms to Label Noise

The core objective of fair FL algorithms, such as DQN-Fed, is to uphold fairness among clients while preserving average accuracy across them. However, it's important to note that these algorithms are not inherently robust against label noise, which refers to instances where data points are mislabeled.

However, by integrating DQN-Fed with label-noise resistant methods from existing literature, we can develop a FL approach that not only ensures fairness among clients but also exhibits robustness against label noise. Specifically, among the label-noise resistant FL algorithms available in the literature, we choose FedCorr (Xu et al., 2022) to be integrated with DQN-Fed. This integration offers a promising avenue for enhancing the performance and resilience of FL models in real-world scenarios affected by label noise.

FedCorr introduces a dimensionality-based filter to identify noisy clients, achieved through the measurement of local intrinsic dimensionality (LID) within local model prediction subspaces. They illustrate the feasibility of distinguishing between clean and noisy datasets by monitoring the behavior of LID scores throughout the training process. For further insights into FedCorr, we defer interested readers to the original paper for a comprehensive discussion.

Following a methodology similar to FedCorr, we utilize a real-world noisy dataset known as Clothing1M. This dataset comprises 1 million clothing images across 14 classes and is characterized by noisy labels, as it is sourced from various online shopping websites, incorporating numerous mislabeled samples.

For our experiments with Clothing1M, we adopt the identical settings as utilized by FedCorr, which are available in their GitHub repository (https://github.com/Xu-Jingyi/FedCorr). Specifically, we employ local SGD with a momentum of 0.5, utilizing a batch size of 16 and conducting five local epochs. Additionally, we set the hyper-parameter $T_1 = 2$ in accordance with their algorithm.

The results are summarized in Table 12. It is evident that the average accuracy achieved by DQN-Fed is approximately 2.2% lower compared to that obtained with FedCorr, indicating DQN-Fed's susceptibility to label noise. However, DQN-Fed demonstrates a notable improvement in ensuring fair client accuracy, aligning with expectations.

Conversely, when DQN-Fed is combined with FedCorr, there is a noticeable enhancement in average accuracy while still preserving satisfactory fairness among clients. This integration showcases the potential of leveraging both methodologies to achieve improved performance and fairness in FL scenarios affected by label noise.

Table 12: Test accuracy on Clothing1M dataset. The reported results are averaged over 5 different seeds.

| Algorithm | $\bar{a}$ | $\sigma_a$ | W(10%) | B(10%) |
|---|---|---|---|---|
| FedAvg | 70.49 | 13.25 | 43.09 | 91.05 |
| FedCorr | 72.55 | 13.27 | 43.12 | 91.15 |
| DQN-Fed | 70.35 | 5.17 | 49.91 | 90.77 |
| FedCorr + DQN-Fed | 72.36 | 8.07 | 46.77 | 91.15 |

## J More on Fairness in FL and ML

### J.1 Sources of unfairness in federated learning

Unfairness in FL can arise from various sources and is a concern that needs to be addressed in FL systems. Here are some of the key reasons for unfairness in FL:

1. **Non-Representative Data Distribution**: Unfairness can occur when the distribution of data across participating devices or clients is non-representative of the overall population. Some devices may have more or less relevant data, leading to biased model updates.

2. **Data Bias**: If the data collected or used by different clients is inherently biased due to the data collection process, it can lead to unfairness. For example, if certain demographic groups are underrepresented in the training data of some clients, the federated model may not perform well for those groups.

3. **Heterogeneous Data Sources**: Federated learning often involves data from a diverse set of sources, including different device types, locations, or user demographics. Variability in data sources can introduce unfairness as the models may not generalize equally well across all sources.

4. **Varying Data Quality**: Data quality can vary among clients, leading to unfairness. Some clients may have noisy or less reliable data, while others may have high-quality data, affecting the model's performance.

5. **Data Sampling**: The way data is sampled and used for local updates can introduce unfairness. If some clients have imbalanced or non-representative data sampling strategies, it can lead to biased model updates (Hamidi et al., 2024b).

6. **Aggregation Bias**: The learned model may exhibit a bias towards devices with larger amounts of data or, if devices are weighted equally, it may favor more commonly occurring devices.

### J.2 Fairness in conventional ML Vs. FL

The concept of fairness is often used to address social biases or performance disparities among different individuals or groups in the machine learning (ML) literature (Barocas et al., 2017). However, in the context of FL, the notion of fairness differs slightly from traditional ML. In FL, fairness primarily pertains to the consistency of performance across various clients. In fact, the difference in the notion of fairness between traditional ML and FL arises from the distinct contexts and challenges of these two settings:

**1. Centralized vs. decentralized data distribution:**

- In traditional ML, data is typically centralized, and fairness is often defined in terms of mitigating biases or disparities within a single, homogeneous dataset. Fairness is evaluated based on how the model treats different individuals or groups within that dataset.

- In FL, data is distributed across multiple decentralized clients or devices. Each client may have its own unique data distribution, and fairness considerations extend to addressing disparities across these clients, ensuring that the federated model provides uniform and equitable performance for all clients.

**2. Client autonomy and data heterogeneity:**

- In FL, clients are autonomous and may have different data sources, labeling processes, and data collection practices. Fairness in this context involves adapting to the heterogeneity and diversity among clients while still achieving equitable outcomes.

- Traditional ML operates under a centralized, unified data schema and is not inherently designed to handle data heterogeneity across sources.

We should note that in certain cases where devices can be naturally clustered into groups with specific attributes, the definition of fairness in FL can be seen as a relaxed version of that in ML, i.e., we optimize for similar but not necessarily identical performance across devices (Li et al., 2019a).

Nevertheless, despite the differences mentioned above, to maintain consistency with the terminology used in the FL literature and the papers we have cited in the main body of this work, we will continue to use the term "fairness" to denote the uniformity of performance across different devices.

### J.3    Fair Algorithms in FL

A seminal method in this domain is agnostic federated learning (AFL) (Mohri et al., 2019). AFL optimizes the global model for the worst-case realization of the weighted combination of user distributions. Their approach involves solving a saddle-point optimization problem, and they employ a fast stochastic optimization algorithm for this purpose. However, AFL exhibits strong performance only for a limited number of clients. In addition, Du et al. (2021) adopted the framework of FedLF and introduced the AgnosticFair algorithm. They linearly parameterized model weights using kernel functions and demonstrated that FedLF can be considered as a specific instance of AgnosticFair. To address the challenges in FedLF, the q-fair Federated Learning (q-FFL) method was introduced by Li et al. (2019a). q-FFL aims to achieve a more uniform test accuracy across users, drawing inspiration from fair resource allocation methods employed in wireless communication networks (Huaizhou et al., 2013). Following this, Li et al. (2020) introduced TERM, a tilted empirical risk minimization algorithm designed to address outliers and class imbalance in statistical estimation procedures. In numerous FL applications, TERM has shown superior performance compared to q-FFL. Adopting a similar concept, Huang et al. (2020b) introduced a method that adjusts device weights based on training accuracy and frequency to promote fairness. Additionally, FCFC (Cui et al., 2021) minimizes the loss of the worst-performing client, effectively presenting a variant of FedLF. Subsequently, Li et al. (2021) introduced Ditto, a multitask personalized FL algorithm. By optimizing a global objective function, Ditto enables local devices to perform additional steps of SGD, within certain constraints, to minimize their individual losses. Ditto proves effective in enhancing testing accuracy among local devices and promoting fairness. Ideas from information theory such as conditional mutual information could also be used to promote fairness in FL (Yang et al., 2023; 2024; Ye et al., 2024).

Our approach shares similarities with *FedMGDA+* (Hu et al., 2022), which treats the FL task as a multi-objective optimization problem. The objective here is to simultaneously minimize the loss function of each FL client. To ensure that the performance of any client is not compromised, *FedMGDA+* leverages Pareto-stationary solutions to identify a common descent direction for all selected clients. In a similar approach, Hamidi & YANG (2024); Mohajer Hamidi & Damen (2024); Hamidi et al. (2025) use ideas from multi-objective optimization to ensure fairness in FL models.

