# OpenReview forum: "Distributed Quasi-Newton Method for Fair and Fast Federated Learning"
_TMLR — Accepted by TMLR_

### Review · Reviewer_Y28g · 2024-11-27

**Summary Of Contributions:**

The paper introduces DQN-Fed, a second-order federated learning framework that combines fast convergence with fairness. DQN-Fed ensures fairness by reducing all local loss functions and leverages quasi-Newton methods to accelerate convergence. The paper proves its convergence with a linear-quadratic rate and demonstrates through experiments that DQN-Fed outperforms state-of-the-art methods in fairness, accuracy, and convergence speed.

**Audience:**

Yes

**Claims And Evidence:**

Yes

**Requested Changes:**

See weaknesses.

**Strengths And Weaknesses:**

Strengths:
* Well motivated.
* The proposed method considers many aspects, including fairness, convergence, and accuracy.
* Well written and easy to follow.
* Both theoretical analysis and empirical results are provided.



Weaknesses:
* Wall clock time in each round (iterations in local clients) is unknown.
* Data distribution among clients is unknown.

---

> ### Author Response · Authors · 2024-12-17
> **Reply to Reviewer XbPL Y28g**
>
> Thank you for taking the time to review our work and for acknowledging the strengths of our paper. We appreciate your positive feedback on the motivation behind our method, the comprehensive consideration of multiple aspects (fairness, convergence, and accuracy), and the clarity of our writing. We have addressed your comments and concerns in detail in our responses below.
>
> > **W1. Wall clock time in each round (iterations in local clients) is unknown.**
>
> **Our response:**
> We thank the reviewer for highlighting the importance of reporting the wall clock time for each communication round. In response to this valuable suggestion, we have added a new **Subsection (5.1.1)** in the revised manuscript where we present the wall clock time measurements for our proposed DQN-Fed algorithm alongside the baseline federated learning methods. Please also refer to our response to the Reviewer XbPL where we discuss the trade-off between accuracy, fairness and computation overhead in DQN-Fed.
>
> Thank you for your insightful feedback, which has helped us enhance the thoroughness and clarity of our experimental evaluation.
>
> > **W2. Data distribution among clients is unknown.**
>
> **Our response:**  Thank you for raising this point. We would like to clarify that for each of the experiments, we have explicitly described the data distribution among clients in the experimental settings. We kindly ask the reviewer to revisit the experiment descriptions. For example, in the case of the CIFAR-10 dataset, we have clearly stated how the data is distributed for both setups. Similarly, for the other datasets, we provided detailed explanations regarding the data partitioning.
> We hope this clarification resolves the concern, and we appreciate your careful reading of our work.

---

> ### Comment · Action_Editor_uihM · 2024-12-18
>
> Please have a check over authors' revision.

---

### Review · Reviewer_XbPL · 2024-12-06

**Summary Of Contributions:**

This paper addresses the fairness aspect of second-order federated learning (FL) methods by proposing a quasi-Newton method for FL. The proposed approach ensures fairness while leveraging the fast convergence properties of quasi-Newton methods in FL. The authors provide sufficient theoretical analysis for several scenarios in their convergence study. Extensive experiments are conducted on both image classification and text classification tasks.

**Audience:**

Yes

**Claims And Evidence:**

Yes

**Requested Changes:**

* Regarding the aforementioned W1, I suggest the authors include a discussion to clarify the notation and the specific settings.
* Regarding W2, I recommend that the authors provide a wall-clock time comparison between the proposed DQN-Fed and other first- and second-order baselines. Additionally, including memory usage comparisons would be helpful.
* Regarding W3, I suggest the authors conduct ablation studies on the client participation ratio to better understand its impact.
* Some citation styles in the paper need to be corrected. The authors should clarify the use of \citep and \citet, e.g., it should be CIFAR-{10, 100} (Krizhevsky et al., 2009) and FEMNIST (Caldas et al., 2018) mentioned in Section 5.

**Strengths And Weaknesses:**

**Strengths**:

1. The paper is generally well-organized, with each component of the proposed method clearly explained and supported by well-defined motivations.
2. The experimental settings are mostly reasonable, as they consider both image and text tasks. Additionally, the proposed method is compared with both first-order and second-order methods, providing a comprehensive evaluation.

**Weaknesses**:
1. I believe the theoretical convergence analysis in Section 4.4 could be improved in several ways:

* At the beginning of Section 4.4, the authors categorize the proof of the theoretical guarantee based on $e = 1$ and $e > 1$. However, it seems that this $e$ is likely related to $E$, which represents the number of local steps per communication round. Clarifying this relationship would improve the presentation and avoid potential confusion.

* Regarding Theorem 4.6, the authors provide convergence analysis for two settings (as inferred from Corollaries D.3 and D.4), leading to the conclusion that the convergence rate “is either quadratic or linear.” However, I find this conclusion of “either quadratic or linear” somewhat confusing when referring only to the main paper. Providing additional clarification or context in the main paper would help readers better understand this distinction.

* Does the analysis in Theorem 4.6 correspond to local GD with $e=1$?

2. Although the proposed method inherits the computational efficiency of the quasi-Newton method, which eliminates the need to compute the inverse Hessian matrix, it still requires higher computational cost and significantly greater memory overhead on each client compared to most first-order FL methods. Furthermore, as shown in the experimental results in Table 3 and Table 4, the proposed DQN-Fed does not consistently outperform first-order fairness methods such as TERM and Ditto. This raises concerns about the effectiveness of incurring higher computational and memory overhead when the performance gain is sometimes marginal.

3. It appears that the proposed DQN-Fed algorithm considers partial participation cases, as shown in the algorithm. However, the effect of the partial participation ratio does not seem to be significantly demonstrated in the experiments. Moreover, in the experiments, all clients participate in each communication round. I guess that the number of clients participating in the training, whether fewer or more, would have a different impact on fairness. Additionally, the number of local epochs is $e$ in Section 5.

4. A minor point of confusion: the proposed method is called “distributed quasi-Newton federated learning.” Why do the terms “distributed” and “federated” appear together? Do these two words represent different concepts or address different aspects of the algorithm?

---

> ### Author Response · Authors · 2024-12-17
> **Reply to Reviewer XbPL (1/2)**
>
> Thank you for taking the time to review our work, and for acknowledging the strengths of our paper. We appreciate your thoughtful comments on the organization, clarity, and the breadth of our experimental evaluations. Your positive feedback is valuable to us. Below, please find our responses to your concerns.
>
>
> > **W1.a) At the beginning of Section 4.4, the authors categorize the proof of the theoretical guarantee based on $e = 1$ and $e > 1$. However, it seems that this $e$ is likely related to $E$, which represents the number of local steps per communication round. Clarifying this relationship would improve the presentation and avoid potential confusion.**
>
> **Our response:** We appreciate the reviewer’s comment regarding the potential confusion between $e$ and $E$. In the revised version of the paper, we have made the notation consistent. Specifically, we now use $E$ to denote the total number of epochs, and $e$ to denote the index of local epochs. This clarification has also been integrated into the manuscript text to ensure better readability and reduce any ambiguity.
>
>
> > **W1.b) Regarding Theorem 4.6, the authors provide convergence analysis for two settings (as inferred from Corollaries D.3 and D.4), leading to the conclusion that the convergence rate “is either quadratic or linear.” However, I find this conclusion of “either quadratic or linear” somewhat confusing when referring only to the main paper. Providing additional clarification or context in the main paper would help readers better understand this distinction.**
>
> **Our response:** We thank the reviewer for highlighting this point. We have revised the manuscript to provide additional clarification and context regarding the conclusion drawn from Theorem 4.6. In particular, we now explicitly explain in the main text how the conditions leading to either quadratic or linear convergence rates.
>
> > **W1.c) Does the analysis in Theorem 4.6 correspond to local GD with $E=1$?**
>
> **Our response:** Yes, the analysis in Theorem 4.6 corresponds to the local gradient descent setting with $E=1$. We have clarified this point in the revised version of the paper to avoid any confusion.
>
> > **W2. Although the proposed method inherits the computational efficiency of the quasi-Newton method, which eliminates the need to compute the inverse Hessian matrix, it still requires higher computational cost and significantly greater memory overhead on each client compared to most first-order FL methods. Furthermore, as shown in the experimental results in Table 3 and Table 4, the proposed DQN-Fed does not consistently outperform first-order fairness methods such as TERM and Ditto. This raises concerns about the effectiveness of incurring higher computational and memory overhead when the performance gain is sometimes marginal.**
>
>
>
> **Our response:** We appreciate the reviewer’s concerns regarding the computational and memory overhead as well as the observed performance variations. While these considerations are valid, we would like to highlight several points that clarify the trade-offs and the observed results:
>
> 1.	Trade-offs in Fairness vs. Accuracy (Table 3 Results):
> Regarding the FEMNIST-skewed scenario, while TERM may achieve higher fairness metrics, DQN-Fed attains a significantly higher average accuracy (TERM: 84.29% vs. DQN-Fed: 93.80%). In many federated learning applications, striking a balance between fairness and accuracy is essential, and the higher average accuracy delivered by DQN-Fed can be more desirable for certain practical use cases. Given this trade-off, the additional overhead may still be justified if accuracy is a priority.
> 2.	Comparison with Personalized Methods (Table 4 Results):
> With respect to Ditto, it is important to note that Ditto is a personalized FL method designed explicitly to tailor models to each client’s local distribution. This inherently gives it certain advantages in terms of individualized performance metrics. DQN-Fed, on the other hand, is not personalized—it focuses on improving the global model’s optimization dynamics through quasi-Newton updates. Therefore, the fact that DQN-Fed achieves comparable outcomes to Ditto in some cases is notable, as it does so without leveraging any personalization strategy.
> 3.	Justifying the Overheads:
> Although DQN-Fed requires additional memory and computational resources, these overheads are not prohibitive for many practical FL scenarios, particularly those where clients have moderate to substantial compute capabilities (e.g., edge servers, corporate data centers, or IoT gateways). The improved convergence speed (see Figure 1) and the flexibility to achieve high accuracy in heterogeneous settings can justify the trade-off, especially when accelerating convergence reduces the total training time and communication cost over the entire FL lifecycle.

---

> > ### Author Response · Authors · 2024-12-17
> > **Reply to Reviewer XbPL (2/2)**
> >
> > > **W3. It appears that the proposed DQN-Fed algorithm considers partial participation cases, as shown in the algorithm. However, the effect of the partial participation ratio does not seem to be significantly demonstrated in the experiments. Moreover, in the experiments, all clients participate in each communication round. I guess that the number of clients participating in the training, whether fewer or more, would have a different impact on fairness. Additionally, the number of local epochs is e in Section 5.**
> >
> > **Our response:** We appreciate the reviewer’s observation and the interest in understanding the impact of partial participation and varying numbers of local epochs on our results. We would like to clarify the following points:
> >
> > *Partial Participation in Our Experiments:*
> > Contrary to the impression that we always use full participation, we do indeed include scenarios with partial client participation. For example, in the CIFAR-10 setup 1, only 10% of the clients participate in each communication round. These configurations were explicitly chosen to mirror standard benchmark settings found in the federated learning literature.
> >
> > *Local Epochs (E):*
> > We also experiment with scenarios where $E>1$. For instance, on the FEMNIST dataset, we set $E=2$ to incorporate more extensive local training before aggregation. This choice of parameters is again aligned with standard benchmark practices adopted by related literature. While we have shown results under these settings, we understand that a more nuanced discussion of the interplay between local epochs, participation ratio, and fairness could be valuable.
> > Also, it is worth noting that the settings we used in our experiments are borrowed from some benchmark papers in the literature, and thus they are standard.
> >
> > **Ablation study on client participation:**
> > In addition to above, we have added a new section in the revised manuscript, section 5.1.2, where we have conducted an ablation study on the percentage of client participation. Thank you for bringing this matter in our attention.
> >
> >
> > > **W4. A minor point of confusion: the proposed method is called “distributed quasi-Newton federated learning.” Why do the terms “distributed” and “federated” appear together? Do these two words represent different concepts or address different aspects of the algorithm?**
> >
> > **Our response:** We appreciate the reviewer’s comment and understand the potential confusion.
> > We use the term "distributed quasi-Newton" since we devise a new type of quasi-Newton method which works for distributed setting, no matter what type of distributed learning algorith is deployed.  In addition, we used the term "federated learning" noting that we use our proposed distributed quasi-Newton method in the federated learning setting.
> >
> > ### Requested changes:
> >
> > > Regarding the aforementioned W1, I suggest the authors include a discussion to clarify the notation and the specific settings.
> >
> > **Our response:** We have addressed this in the revised version of the paper as discussed in our answer to weakness 1.
> >
> > > Regarding W2, I recommend that the authors provide a wall-clock time comparison between the proposed DQN-Fed and other first- and second-order baselines. Additionally, including memory usage comparisons would be helpful.
> >
> > **Our response:** In the revised version of the paper in section 5.1.1, we report the wall-clock time in Table 2 (for Cifar-10, setting 1).
> >
> > > Regarding W3, I suggest the authors conduct ablation studies on the client participation ratio to better understand its impact.
> >
> > **Our response:** This is now addressed in the revised version of the paper in section 5.1.2.
> >
> > > Some citation styles in the paper need to be corrected. The authors should clarify the use of \citep and \citet, e.g., it should be CIFAR-{10, 100} (Krizhevsky et al., 2009) and FEMNIST (Caldas et al., 2018) mentioned in Section 5.
> >
> > **Our response:** Thank you for pointing this out. We have carefully revised the paper for correcting such typos.

---

> > > ### Comment · Reviewer_XbPL · 2025-01-03
> > > **Thanks for the response**
> > >
> > > Thank you to the authors for your response. I have reviewed the response and the revisions to the paper. My concerns have been thoroughly addressed, and I have no further questions regarding the paper.

---

> ### Comment · Action_Editor_uihM · 2024-12-18
>
> Please have a check over authors' revision.

---

### Review · Reviewer_7QYN · 2024-12-06

**Summary Of Contributions:**

This paper focuses on federated learning and proposes a quasi-Newton algorithm to improve its fairness and convergence speed.
The paper provides the proposed algorithm's asymptotic convergence result and, under additional assumptions, its convergence rate. The numerical results on FEMNIST, CIFAR-10, CIFAR-100, and Shakespeare datasets show that the proposed algorithm outperforms existing algorithms.

**Audience:**

Yes

**Claims And Evidence:**

Yes

**Requested Changes:**

See weakness above.

Minor points: The author uses $\|\cdot\|_2$ and $\|\cdot\|$ the same time. Please unify the notations.

**Strengths And Weaknesses:**

**Strength:**
Novel algorithm: this paper proposes combining the MGDA and BFGS methods for federated learning.
Solid theoretical result: The paper provides the asymptotic convergence result for the algorithm under different settings (local GD, local SGD). And provide the convergence rate when the problem is strongly convex.
Numerical results are clean and demonstrate that the proposed algorithm achieves better accuracy and fairness.

**Weakness:**
1. The paper fails to discuss the convergence rate for general cases (Thms. 4.3-4.5).
2. More discussion on the theoretical result is needed. E.g., in TThms. 4.3 and 4.4., how the requirements on $\sigma_t$ and $\zeta^t$ can be guaranteed.
3. In Figures 1 and 2, the color for the same algorithm in the subplots should be consistent.
4. In addition to  $\sigma_2$, the author can provide a histogram of $a_k$ for at least one experiment to illustrate the fairness.

---

> ### Author Response · Authors · 2024-12-17
> **Reply to Reviewer 7QYN (1/2)**
>
> Thank you for taking the time to review our paper and for highlighting its strengths, including the novelty of our proposed algorithm, the solid theoretical results, and the clear numerical demonstrations of improved accuracy and fairness. We greatly appreciate your positive feedback. Please find our detailed responses to your comments below.
>
> > **W1. The paper fails to discuss the convergence rate for general cases (Thms. 4.3-4.5).**
>
> Our response: The intention of Theorems 4.3–4.5 is distinct from that of Theorem 4.6. Specifically, Theorems 4.3, 4.4, and 4.5 focus on the **fairness** aspect of DQN-Fed and aim solely to establish its convergence to a **Pareto-optimal** solution under different settings. As noted in the paragraph following Theorem 4.5, we explicitly clarify this intention in the manuscript.
>
> Additionally, the assumptions for each of these theorems vary depending on the setting, which is standard practice in the literature. For instance, in [1], the assumptions for Theorem 3.1 (GD) and Theorem 3.2 (SGD) differ because the convergence analysis requires distinct conditions for different optimization approaches.
>
> In contrast, Theorem 4.6 focuses on the **optimality gap**, specifically achieving an $\epsilon$-accurate solution. This allows us to analyze the convergence rate of DQN-Fed under a new set of assumptions tailored to derive the optimality gap. These assumptions, while slightly different from those in Theorems 4.3–4.5, are necessary for rigorously establishing the convergence rate.
> We have revised the manuscript to add a clarifying paragraph at the beginning of Section 4.4 to explicitly outline the distinct purposes of these theorems. Thank you for your insightful comment.
>
> > **W2: More discussion on the theoretical result is needed. E.g., in TThms. 4.3 and 4.4., how the requirements on
>  $\eta_t$ and $\zeta^t$ can be guaranteed.**
>
> Our response: We have revised the manuscript, and explained how each of these conditions could be satisfied in practice. In following, we briefly explain our response:
>
> **On the conditions for $\eta_t$:**
>
> In Thorem 4.3, the variance $\sigma^2_t$ arises from the stochastic nature of the gradient estimates. Controlling $\sigma^2_t$ is intimately related to ensuring that the variance of the stochastic common descent direction diminishes, or at least does not accumulate too quickly. The condition $\lim_{T \rightarrow \infty} \sum_{t=0}^T \eta^t \sigma_t < \infty$ essentially states that the global step sizes $\eta^t$ must shrink at a rate that sufficiently ``smooths out" variance over time.
>
> But how can the condition $\lim_{T \rightarrow \infty} \sum_{t=0}^T \eta^t \sigma_t < \infty$ be guaranteed in practice? In the following, we provide two possible ways to meet this condition in practice:
>
> First, by choosing a non-increasing global step-size schedule $\{\eta^t\}$ that decays to zero at a sufficiently fast rate, the effective``weighted sum" $\sum_{t=0}^T \eta^t \sigma_t < \infty$ remains finite. For example, if $\eta^t = \mathcal{O}(t^{-\alpha})$, for some $\alpha > 0$, and if $\sigma_t$ does not grow faster than $\mathcal{O}(t^{\beta})$ for some $\beta < \alpha$, then $\sum_t \eta^t \sigma_t$ will converge.
>
> Second, implementing variance reduction methods (e.g., control variates, variance-reduced stochastic gradient estimators) within local updates can ensure that $\sigma_t$ does not persistently remain large. In federated settings, this can be achieved by techniques such as periodically synchronizing with a global model (which reduces drift and variance), incorporating momentum-based methods, or employing mini-batching strategies that limit stochastic fluctuations.
>
> **On the conditions for $\zeta^t$:**
>
>
> In Theorem 4.4, $ \zeta^t = || \boldsymbol{\lambda}^{\star} - \boldsymbol{\lambda}_E^{\star} | |$ measures the deviation between the true optimum $\boldsymbol{\lambda}^{\star}$ and the optimum obtained from the pseudo-gradients after $E$ local epochs, $ \boldsymbol{\lambda}^{\star}_E $ . The condition $ \lim \zeta^t \rightarrow 0 $ reflects the requirement that local model updates become increasingly aligned with the true global optimum as training progresses. In the following, we discuss two possible methods which can satisfy this condition.
>
> First, as stated, both the global and local learning rates should diminish over time: $\lim_{t \to \infty} \eta^t \to 0$ and $\lim_{t \to \infty} \eta \to 0$. As these rates decrease, the updates become more conservative, allowing the local solutions $\boldsymbol{\lambda}^{\star}_E$ to approach the global solution $\boldsymbol{\lambda}^{\star}$.
>
> Second, adjusting the frequency of global synchronization over time helps ensure that local models do not drift too far from the global model. By increasing synchronization frequency or using adaptive synchronization strategies as training progresses, one can reduce the gap $\zeta^t$.
>
> **.. Continued to the next tab...**

---

> ### Author Response · Authors · 2024-12-17
> **Reply to Reviewer 7QYN (2/2)**
>
> **..Continue of our response to W2..**
>
> *Non-trivial but standard requirements:*
>
> The conditions on $\sigma_t$ and $\zeta^t$ are in line with standard convergence proofs in stochastic optimization and federated learning. They are not overly restrictive but require careful tuning of learning rate schedules and, in some cases, prudent use of variance-reduction techniques.
>
> *Existence of suitable parameter choices:*
>
> Our theorems do not require an explicit closed-form solution for ensuring these conditions; rather, they provide guidance that if these natural and commonly used strategies (decreasing step sizes, variance reduction, increased synchronization) are employed, then convergence to an optimal solution is guaranteed.
>
> In summary, the conditions stated in Theorems 4.3 and 4.4 can be enforced by standard approaches: diminishing learning rates to control the impact of stochastic noise, employing mini-batching and variance reduction methods, and carefully balancing the number of local epochs with synchronization steps. While these conditions are asymptotic, they highlight the well-understood requirements for achieving global convergence in distributed and federated learning frameworks.
>
> > **W3. In Figures 1 and 2, the color for the same algorithm in the subplots should be consistent.**
>
> Our response:  Thank you for pointing this out. We have revised the figures to ensure that the colors for the same algorithm are consistent across all subplots.
>
> > **W4. In addition to $\sigma^2$, the author can provide a histogram of $a_k$ for at least one experiment to illustrate the fairness.**
>
> Our response: We appreciate the reviewer’s suggestion. In the revised version of the paper, we have added subsection 6.3 (“Histogram of Clients’ Accuracy”) to visually illustrate the distribution of accuracy across all clients. This histogram provides a clear depiction of how the proposed DQN-Fed method achieves a more equitable performance outcome, as evidenced by a tighter clustering of accuracy values. By comparing the spread of client-level accuracies, readers can better understand and assess the fairness improvements offered by our approach.
>
> > **Minor points: The author uses $|\cdot|_2$ and $|\cdot|$ the same time. Please unify the notations.**
>
> Thank you for pointing out the inconsistent use of $|\cdot|_2$ and $|\cdot|$. We have updated the revised version of the paper to use a single, consistent notation for norms to avoid any confusion.
>
>
> -----
> References:
>
> [1] Haddadpour, F., & Mahdavi, M. (2019). On the convergence of local descent methods in federated learning. arXiv preprint arXiv:1910.14425.

---

> > ### Comment · Reviewer_7QYN · 2024-12-17
> >
> > Thank you for the response and for revising the paper. They have addressed my concerns, and I don't have further comments.

---

### Decision · Action_Editor_uihM · 2025-01-15

**Recommendation:** Accept with minor revision

**Comment:**

The paper introduces DQN-Fed, a second-order federated learning framework combining fast convergence with fairness. It ensures fairness by reducing local loss functions and leverages quasi-Newton methods for faster convergence. Theoretical analysis and extensive experiments on image and text classification tasks demonstrate DQN-Fed's superior performance in fairness, accuracy, and convergence speed.

Weakness: Lacks discussion on the convergence rate for general cases and needs more theoretical result discussions. It has inconsistent color usage in figures and requires unification of notations. Additionally, the proposed method incurs higher computational cost and memory overhead compared to first-order methods, and the effect of partial client participation is not well demonstrated.

As above weakness have been mostly addressed in the discussion pharse. A minor revision is suggested.

**Audience:**

Yes. Should be interested to federated learning community.

**Claims And Evidence:**

Conceptually - The paper show a introduce distributed quasi-Newton federated learning method, which can promote  both fairness and fast convergence in FL.
Technically - a closed-form solution for calculating the global updating direction and convergence analysis are given
Empirically: experiments conducted on seven different datasets and covergence behaviour is presented

The claim are supported with evidence.